# Nr6a1 controls *Hox* expression dynamics and is a master regulator of vertebrate trunk development

Yi-Cheng Chang[1,2], Jan Manent[1,2], Jan Schroeder [2,3,4], Siew Fen Lisa Wong[1,2], Gabriel M. Hauswirth [1,2], Natalia A. Shylo[5], Emma L. Moore [5], Annita Achilleos [5,6], Victoria Garside[1,2], Jose M. Polo [2,3,4], Paul Trainor [5,7] & Edwina McGlinn [1,2] ✉

The vertebrate main-body axis is laid down during embryonic stages in an anterior-to-posterior (head-to-tail) direction, driven and supplied by posteriorly located progenitors. Whilst posterior expansion and segmentation appears broadly uniform along the axis, there is developmental and evolutionary support for at least two discrete modules controlling processes within different axial regions: a trunk and a tail module. Here, we identify Nuclear receptor subfamily 6 group A member 1 (Nr6a1) as a master regulator of trunk development in the mouse. Specifically, Nr6a1 was found to control vertebral number and segmentation of the trunk region, autonomously from other axial regions. Moreover, Nr6a1 was essential for the timely progression of *Hox* signatures, and neural versus mesodermal cell fate choice, within axial progenitors. Collectively, Nr6a1 has an axially-restricted role in all major cellular and tissue-level events required for vertebral column formation, supporting the view that changes in Nr6a1 levels may underlie evolutionary changes in axial formulae.

Vertebrate animals exhibit great diversity in their axial formulae, that is, the number and identity of elements that constitute the vertebral column. Axial formulae are established at early developmental stages when regional identity is superimposed on the continual process of somitogenesis[1]. During primary body formation, presacral vertebrae arise from dual-fated progenitor cells known as neuromesodermal progenitors (NMPs) that reside in the epiblast[2,3], with the process of gastrulation providing a constant supply of descendant cells directly to the posterior presomitic mesoderm (PSM). The resulting PSM expansion is balanced by the periodic budding of tissue from the anterior PSM, generating vertebrae precursors known as somites. As gastrulation terminates, a subset of NMPs relocate internally to the chordoneural hinge of the tailbud, where they continue to supply descendant cells destined to form post-sacral vertebrae during secondary body formation[4,5]. This continues until progenitor exhaustion/PSM decline halts elongation and defines total vertebral number for that species[6–8].

While each newly formed somite appears seemingly identical along the entire anterior-to-posterior (A-P) axis, the positional information that instructs regionally-appropriate vertebra morphology is already present prior to somite scission[9,10]. Central orchestrators of this positional information are the homeodomain-containing Hox proteins, a series of 39 transcription factors (in mouse/human) that work within a complex spatio-temporal hierarchy to pattern the

[1]EMBL Australia, Monash University, Clayton, Victoria 3800, Australia. [2]Australian Regenerative Medicine Institute, Monash University, Clayton, VIC 3800, Australia. [3]Department of Anatomy and Developmental Biology, Monash University, Clayton, VIC, Australia. [4]Development and Stem Cells Program, Monash Biomedicine Discovery Institute, Clayton, VIC, Australia. [5]Stowers Institute for Medical Research, Kansas City, Missouri, USA. [6]University of Nicosia, Nicosia, Cyprus. [7]Department of Anatomy and Cell Biology, University of Kansas Medical Center, Kansas City, Kansas, USA. ✉e-mail: edwina.mcglinn@monash.edu

vertebrae[11]. Specifically, it is the temporally-ordered activation of *Hox* gene expression within axial progenitors[12], reinforced by their genomic clustering at 4 distinct loci[13], that results in a spatially overlapping "Hox-code" along the A-P axis that governs regional identity[11].

Evolutionary changes in vertebral number that selectively alter either the trunk or the tail/caudal regions have been repeatedly observed across the vertebrates[14–17], supporting the view that axial elongation is driven by consecutive developmental modules that can be manipulated independently. At a cellular level, the trunk-to-tail (T-to-T) transition is marked by a switch in axial progenitor source from epiblast- to tailbud-derived[8,18], and is coincident with a shift from an expanding to a depleting PSM size[6]. Transplantation experiments have shown that these T-to-T changes in cellular state and activity are not irreversible[4], demonstrating that intrinsic exhaustion of axial progenitors does not follow a timer mechanism and that axial identity is able to be reset to a more anterior fate based on environmental cues[4,19]. These data, together with the observation of a temporally changing transcriptomic signature of axial progenitors[20,21], as opposed to a stable signature, suggest an evolving developmental program as the main-body axis is laid down but one that can be broadly delineated into trunk and tail modules.

Gdf11 is one of the most prominent factors known to control the correct timing of the T-to-T transition. Genetic deletion of *Gdf11*[22] or its receptors *Acvr2a* and *Acvr2b*[23,24] in the mouse significantly expands the trunk region, while constitutive activation of this signalling pathway had the opposite effect[25]. However, Gdf11 signalling has pleiotropic effects across the vertebral column, with patterning changes as far anterior as the cervical region as well as tail truncation observed in *Gdf11*[−/−] mice[22]. It is currently unclear whether the cell-intrinsic mechanisms downstream of Gdf11 signalling are common across all phenotypes, though axially-restricted effectors such as Pou5f1/Oct4 in the trunk[26], and the Lin28-*let7* axis in the tail[27–29] have been identified.

Nuclear receptor subfamily 6 group A member 1 (Nr6a1; previously called GCNF) is an orphan-nuclear receptor whose genetic deletion in the mouse results in early embryonic lethality[30]. *Nr6a1* is expressed widely within the early mouse embryo[30,31], and is one of a select suite of marker genes that exhibits temporally-restricted expression within NMPs at embryonic day (E)8.5[20]. Of particular note, an activating single nucleotide polymorphism (SNP) within *Nr6a1* has been associated with increased trunk vertebral number in the domesticated pig[32–35], a trait that has been selected for increased meat production[36], with similar SNPs now found in domesticated sheep[37]. In mouse and other vertebrate species, the *Nr6a1* 3′-untranslated region harbours multiple binding sites for miR-196 and let-7[38–40], microRNA families known to constrain trunk[38] and tail[27] vertebral number respectively, suggesting the potential for Nr6a1 to function during vertebral column formation in an axially-restricted manner, though to date, this remains to be tested.

Here, we identify Nr6a1 as a master regulator of trunk elongation, segmentation, patterning and lineage allocation in the mouse. *Nr6a1* expression within axial progenitors is dynamic, being positively reinforced by Wnt signalling at early stages and sharply terminated by the combined actions of Gdf11 and miR-196 at the T-to-T transition. Nr6a1 acts in a dose-dependent manner to control the number of thoraco-lumbar elements in a meristic not homeotic manner, with the subsequent termination of *Nr6a1* expression essential for tail development. Furthermore, Nr6a1 controls the timely progression of *Hox* expression derived from all Hox clusters. Specifically, Nr6a1 activity enhances the expression of several trunk *Hox* genes whilst temporally constraining the expression of posterior *Hox* genes. Collectively, our data reinforce the view that axial elongation is controlled by at least two developmental modules, the first being Nr6a1-dependent and the second Nr6a1-independent, with Nr6a1 providing central cross-talk between elongation and patterning.

## Results

### Expression of *Nr6a1* correlates with trunk formation in the mouse

The genomic locus encompassing *Nr6a1* produces multiple transcripts in the mouse (Supplementary Fig. 1A). These include two largely overlapping *Nr6a1* sense transcripts that produce proteins with identical DNA-binding and ligand-binding domains (Supplementary Fig. 1B), with a common 3′untranslated region harbouring multiple microRNA (miRNA) binding sites (Supplementary Fig. 1C). Within intron 3 of the long isoform, multiple antisense transcripts were identified including a long non-coding RNA (*Nr6a1os*) and a polycistronic transcript that encodes the *miR-181a2* and *miR-181b-2* microRNAs (miRNAs).

Whole mount in situ hybridisation utilising a riboprobe that bound both protein-coding sense transcripts revealed widespread *Nr6a1* expression at embryonic day (E) 8.5 (Fig. 1A), including the posterior growth zone which houses various progenitor populations required for axial elongation[8]. Interrogation of a published single-cell RNA-seq dataset produced from E6.5 and E8.5 embryos[41] confirmed robust expression of *Nr6a1* at these early stages in NMPs, caudal mesoderm and the caudolateral epiblast (Fig. 1B), with lower expression in many other progenitor and mature populations where it has the potential to function. At E9.5, widespread *Nr6a1* expression was largely maintained consistent with previous studies[42,43], though with notable absence of expression in the heart and a visible clearing of expression from the posterior growth zone during the period known as the T-to-T transition (Fig. 1A). This temporal decline in *Nr6a1* within E9.5 NMPs is supported at the single-cell level[20]. At E10.5, restricted expression became apparent within the mid- and hind-brain, cranial neural crest cells, otic vesicle and dorsal root ganglia of the trunk, with a complete absence of expression caudal to the last-formed somite. By E12.5, *Nr6a1* expression was barely detectable throughout the embryo, consistent with a "mid-gestation" pattern of expression[39].

The rapid clearance of *Nr6a1* from the tailbud around E9.5 was striking and suggested the posterior limit of its expression may be controlled by factors that position the T-to-T transition. Indeed, *Nr6a1* expression levels are known to be moderately repressed in vivo by *miR-196*, a factor that constrains thoraco-lumbar (T-L) number by 1[38]. To dissect *Nr6a1* regulatory mechanisms at this posterior boundary further, we took advantage of a series of recently generated mouse mutant lines which harbour an increasing number of thoraco-lumbar vertebrae dependent on the individual and combinatorial deletion of *miR-196* and *Gdf11* alleles[44]. Quantification of *Nr6a1* expression within E9.5 tailbud tissue from across this allelic series revealed an overall positive correlation between early *Nr6a1* expression level and the number of additional T-L vertebrae that form later in development (Fig. 1C). This increase in *Nr6a1* expression relative to wildtype persisted at E10.5 and in extreme genotypes, where 8 or 13 additional T-L elements ultimately form, this differential expression escalated even further.

Spatial analysis of *Nr6a1* in E10.5 *Gdf11*[−/−] embryos confirmed persistent expression in the caudal neural tube and mesoderm expanding toward the embryonic tip (Fig. 1D), reminiscent of ectopic *Lin28a* and *Lin28b* expression observed in embryos of the same genotype[28]. Collectively, these data revealed a dynamic pattern of *Nr6a1* expression within tailbud tissue over time, with expression terminated by the synergistic action of *miR-196* and Gdf11 signalling.

### Nr6a1 supports trunk elongation and inhibits sacro-caudal identity

To provide insights into the mechanistic role of Nr6a1 in axial elongation, we first performed microarray transcriptomic analyses on E9.0-9.5 wildtype and *Nr6a1*[−/−] mutant embryos. The top 50 genes downregulated in *Nr6a1*[−/−] mutants compared to wildtype were analysed by

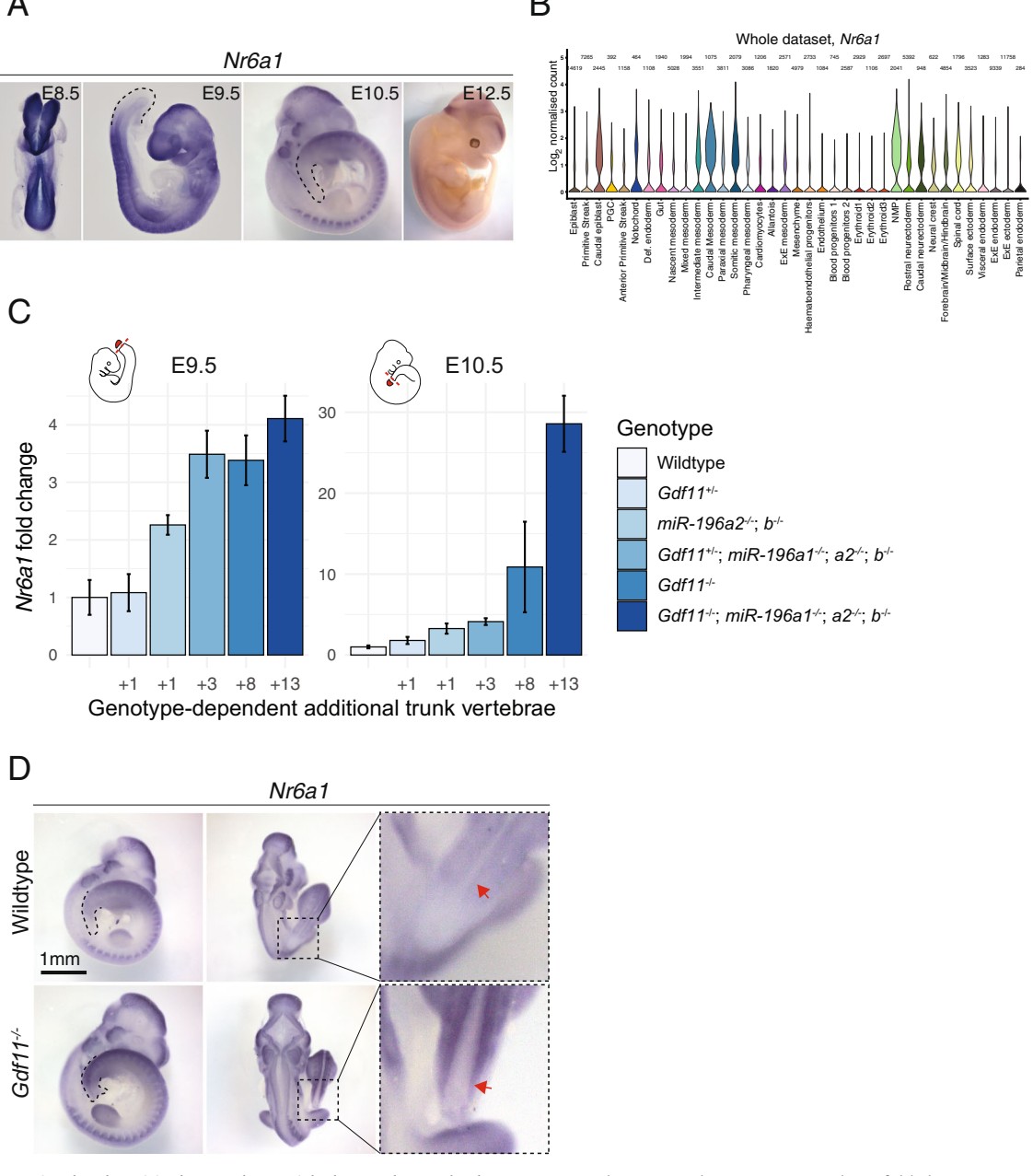

**Fig. 1 | *Nr6a1* expression level positively correlates with the trunk vertebral number and is regulated by known mediators of the trunk-to-tail transition.** **A** *Nr6a1* is expressed broadly at embryonic day (E)8.5, with expression becoming progressively excluded from the posterior growth zone starting E9.5. By E10.5 *Nr6a1* exhibits tissue-specific expression, followed by complete tissue clearance by E12.5. **B** Violin plot illustrates the Log₂ normalised count, which was derived from the raw counts from each cells divided by their size factors, of *Nr6a1* in cells of the E6.5 to E8.5 mouse embryo, revealing robust expression within neuromesodermal progenitors (NMP), caudal mesoderm and caudolateral epiblast. Data derived from ref. [41]. **C** Expression level of *Nr6a1* within the tailbud positively correlates with the number of trunk vertebrae that form later in development. Quantitative PCR analysis of *Nr6a1* within tailbud tissue isolated from *Gdf11* and *miR-196* single and compound mutant embryos, represented as a fold change compared to wildtype (defined as 1). The number of biologically independent samples assessed: WT E9.5 $n = 3$, E10.5 $n = 4$; *Gdf11⁺/⁻* E9.5 $n = 3$, E10.5 $n = 3$; *miR-196a2⁻/⁻;b⁻/⁻* E9.5 $n = 3$, E10.5 $n = 4$; *Gdf11⁺/⁻; miR-196a1⁻/⁻;a2⁻/⁻;b⁻/⁻* E9.5 $n = 3$, E10.5 $n = 4$; *Gdf11⁻/⁻* E9.5 $n = 3$, E10.5 $n = 2$; *Gdf11⁻/⁻; miR-196a1⁻/⁻;a2⁻/⁻;b⁻/⁻* E9.5 $n = 3$, E10.5 $n = 2$; Error bar represents standard deviation. Source data are provided as a Source Data file. The number of additional thoraco-lumbar vertebrae observed across the allelic deletion series[44] is indicated. **D** Whole mount in situ hybridisation reveals that the quantitative increase of *Nr6a1* in E10.5 *Gdf11⁻/⁻* tailbud tissue relative to wildtype reflects a lack of timely clearance from the posterior growth zone. Consistent spatial changes were observed in 2 biologically independent samples/genotype. Red arrow indicates the neural tube.

ToppGene, which predicted that regionalisation, pattern specification, skeletal system development and morphogenesis, and anterior-posterior patterning were key biological processes disrupted in *Nr6a1⁻/⁻* mutants (Supplementary Data 1 and Supplementary Fig. 2A). ToppGene analysis also predicted multiple phenotypes associated with abnormal morphology of the axial skeleton as likely outcomes resulting from Nr6a1 loss-of-function (Supplementary Table 1). Lastly, DAVID analysis revealed that the protein families over-represented in the top 50 downregulated genes in *Nr6a1⁻/⁻* mutants included Homeobox proteins and Homeodomain-like proteins (Supplementary Table 1) suggesting a putative link between Nr6a1 and Hox-dependent patterning mechanisms.

The early lethality of $Nr6a1^{-/-}$ embryos complicates evaluation of its role during vertebral column formation[30,45]. To circumvent this, we next generated a ubiquitous but temporally-controlled conditional knockout model, crossing the $Nr6a1^{flx/flx}$ line[46] with the $CMV$-$Cre^{ERT2}$ deleter line[47]. Low dose Tamoxifen (Tam) was administered on a single day of development from E6.25-E9.25 and skeletal analysis performed at E18.5, revealing a reduction in the number of thoracic elements in $CMV$-$Cre^{ERT2}$; $Nr6a1^{flx/flx}$ embryos when Tam was administered at or before E8.25 (Supplementary Fig. 2B, C). With the time-window of action defined, we next deleted $Nr6a1$ specifically within axial progenitors using the $TCre^{ERT2}$ deleter line[48]. Following administration of Tam at E7.5, a dose-dependent reduction in total vertebral number (TVN) and widespread vertebral patterning changes could be observed (Fig. 2A, B and Supplementary Fig. 3A). Compared to $TCre^{ERT2}$-positive control embryos, heterozygous conditional deletion of $Nr6a1$ resulted in 2 less T elements (Fig. 2A) and an overall reduction in TVN by 1 (Supplementary Fig. 3A) which we suggest stems from serial posteriorising homeotic transformations from the lower thoracic region onwards. For example, while the normal complement of 7 sternal rib attachments were unchanged in $TCre^{ERT2}$;$Nr6a1^{+/flx}$ embryos, the transitional vertebra (vertebra 17 (T10) in controls) was anteriorly displaced by one and all elements from the first lumbar vertebra onwards were anteriorly displaced by two. This phenotype was enhanced following homozygous conditional deletion of $Nr6a1$, with the number of sternal rib attachments reduced to 5, an overall loss of 4 T elements and a reduction in TVN by 3 (Fig. 2A and Supplementary Fig. 3A).

Removal of the pelvic bones in a WT embryo allowed visualisation of the normal winged-shaped transverse process on sacral elements S1 and S2 which at this stage are becoming fused laterally, and the rostrally-pointing transverse process of lumbar elements L4, L5, L6 (Fig. 2B). In $TCre^{ERT2}$;$Nr6a1^{flx/flx}$ embryos, positioning of the normal (albeit malformed) sacral region was appropriately spaced relative to the last formed thoracic element, defined by the presence of adjacent rudimentary pelvic/hindlimb bones (blue S, Fig. 2B). Rostral to this, the transverse processes of all intervening post-thoracic vertebral elements assume a flattened morphology which, in numerous embryos, show lateral fusion between adjacent elements (red S, Fig. 2B; asterisk marks lateral fusions). These unique characteristics of sacral morphology indicated an almost complete transformation of lumbar to sacral identity in $TCre^{ERT2}$;$Nr6a1^{flx/flx}$ embryos.

To corroborate this morphological assessment, we performed whole-mount in situ hybridisation for $Hoxd11$, a known regulator of sacral identity[49]. At E10.5, we see a striking shift in the rostral boundary of $Hoxd11$ in $TCre^{ERT2}$;$Nr6a1^{flx/flx}$ embryos compared to somite-matched WT embryos (Fig. 2D). This shift was specific to the paraxial mesoderm, with ectopic $Hoxd11$ expression observed in at least 6-7 somites rostral to the normal boundary of expression, correlating well with the extent of skeletal transformations observed (Fig. 2B). We also assessed the expression of retinoic acid signalling components $Aldh1a2$ and $Cyp26a1$ at E9.5 and E10.5 but see no major spatio-temporal differences between genotypes (Supplementary Fig. 4) that prefigure the rostral shift in $Hox$ expression or that may underlie body plan changes. Collectively, these results demonstrated an essential and dose-dependent role for Nr6a1 in the appropriate timing of patterning events during axial elongation, with the precocious activation of more posterior programs following Nr6a1 depletion predicted to temporally advance termination of elongation[28,44,50], leading to the reduction in TVN observed.

In addition to these patterning and numerical changes, homozygous conditional deletion of Nr6a1 resulted in rib fusion defects and vertebral malformations throughout the entire thoracic and extended-sacral regions (Fig. 2A and Supplementary Fig. 5). Surprisingly, however, vertebral morphology immediately after the hindlimb reverted back to normal, with tail elements of $TCre^{ERT2}$;$Nr6a1^{flx/flx}$ embryos indistinguishable from controls (Fig. 3A). Consistent with this, characterisation of the early somite marker $Uncx4.1$ at E10.5 revealed highly dysmorphic somite formation in the interlimb and hindlimb regions of $TCre^{ERT2}$;$Nr6a1^{flx/flx}$ embryos compared to controls, with a sharp switch back to normal appearance immediately caudal to the hindlimb bud (Fig. 3B). Collectively, these results demonstrate an axially-restricted function for Nr6a1 in terms of vertebral number, patterning and morphology, reinforcing the view that distinct programs govern the trunk and the tail regions - the former being Nr6a1-dependent and the latter Nr6a1-independent.

## Nr6a1 controls timely posterior $Hox$ expression in the tailbud

The sequential activation of $Hox$ genes in time, and subsequently in space, underpins body patterning along the vertebrate A-P axis[12,49,51]. Given the rostral shift in $Hoxd11$ expression within maturing somites following conditional deletion of Nr6a1 (Fig. 2D), we next wanted to address whether this was secondary to the precious temporal activation of this gene, and potentially other $Hox10$-$13$ paralog genes, within the E9.5 tailbud tissue. Tissue was collected from somite-matched (+/−1) embryos (Supplementary Fig. 6) and the expression of several posterior $Hox$ genes found to be significantly upregulated in $TCre^{ERT2}$;$Nr6a1^{flx/flx}$ tailbuds compared to controls, including $Hoxc12$, $Hoxc12$, $Hoxb13$, $Hoxc13$ and $Hoxd13$, with others following a similar trend (Fig. 4A). For many of these genes, a dose-dependent control of $Hox$ expression levels by Nr6a1 is suggested, although not significant with current sample numbers. The most highly upregulated $Hox$ gene amongst this list was $Hoxb13$, increasing more than 200-fold in $TCre^{ERT2}$;$Nr6a1^{flx/flx}$ tailbuds (Fig. 4A).

Spatial analysis of $Hoxb13$ in wildtype E9.5 embryos (Fig. 4B) confirmed expression was limited to endodermal cells of the hindgut at this stage[52]. In contrast, E9.5 $TCre^{ERT2}$;$Nr6a1^{flx/flx}$ embryos showed additional ectopic expression in the tailbud mesoderm (Fig. 4B). Ectopic $Hoxb13$ expression persisted in E10.5 $TCreER^{T2}$;$Nr6a1^{flx/flx}$ embryos throughout the tailbud mesoderm, and increased $Hoxb13$ expression levels were observed in the neural tube relative to wildtype (Fig. 4B). These data reveal an essential role for Nr6a1 in ensuring correct spatio-temporal activation of a collective posterior $Hox$ code within the tailbud and, specifically, in preventing their precocious expression at stages prior to the T-to-T transition.

## Nr6a1 clearance is essential for the trunk-to-tail transition

During development, the clearance of expression signatures can be instructive or passive. To test the morphological and molecular importance of timely Nr6a1 clearance at the T-to-T transition, we next turned to a gain-of-function approach where transgenic expression of Nr6a1 downstream of $Cdx2$ regulatory elements[53] maintained gene expression in the tailbud throughout axial elongation ($Cdx2P$:$Nr6a1$). Skeletal analysis of $Cdx2P$:$Nr6a1$ embryos at E18.5 revealed thoraco-lumbar expansion, constituting an additional 2T and 2-4L vertebrae relative to wildtype (Fig. 5A). As the number of sternal rib attachments was unaltered in these transgenics (Supplementary Fig. 7A), the additional T elements are considered to be of caudal thoracic identity. Within the $Cdx2P$:$Nr6a1$ extended lumbar region, almost all of the elements harboured an accessory process known as the anapophysis, a feature usually restricted to the first 3 lumbar elements in wildtype (Fig. 5B). As such, we can pinpoint the presacral expansion in $Cdx2P$:$Nr6a1$ embryos to elements immediately surrounding the T-L junction, in terms of their identity.

Increased Nr6a1 levels did not overtly affect vertebrae morphology within the T−L region up until the most caudal lumbar elements, after which, all sacro-caudal elements became small, fused and exhibited defects in dorsal closure (Fig. 5A and Supplementary Fig. 7B). These regionally-restricted phenotypic changes were complementary to those observed following conditional deletion of Nr6a1 (Fig. 2A), and demonstrate that the normal clearance of Nr6a1 at the T-to-T transition is not an inconsequential outcome of an altered

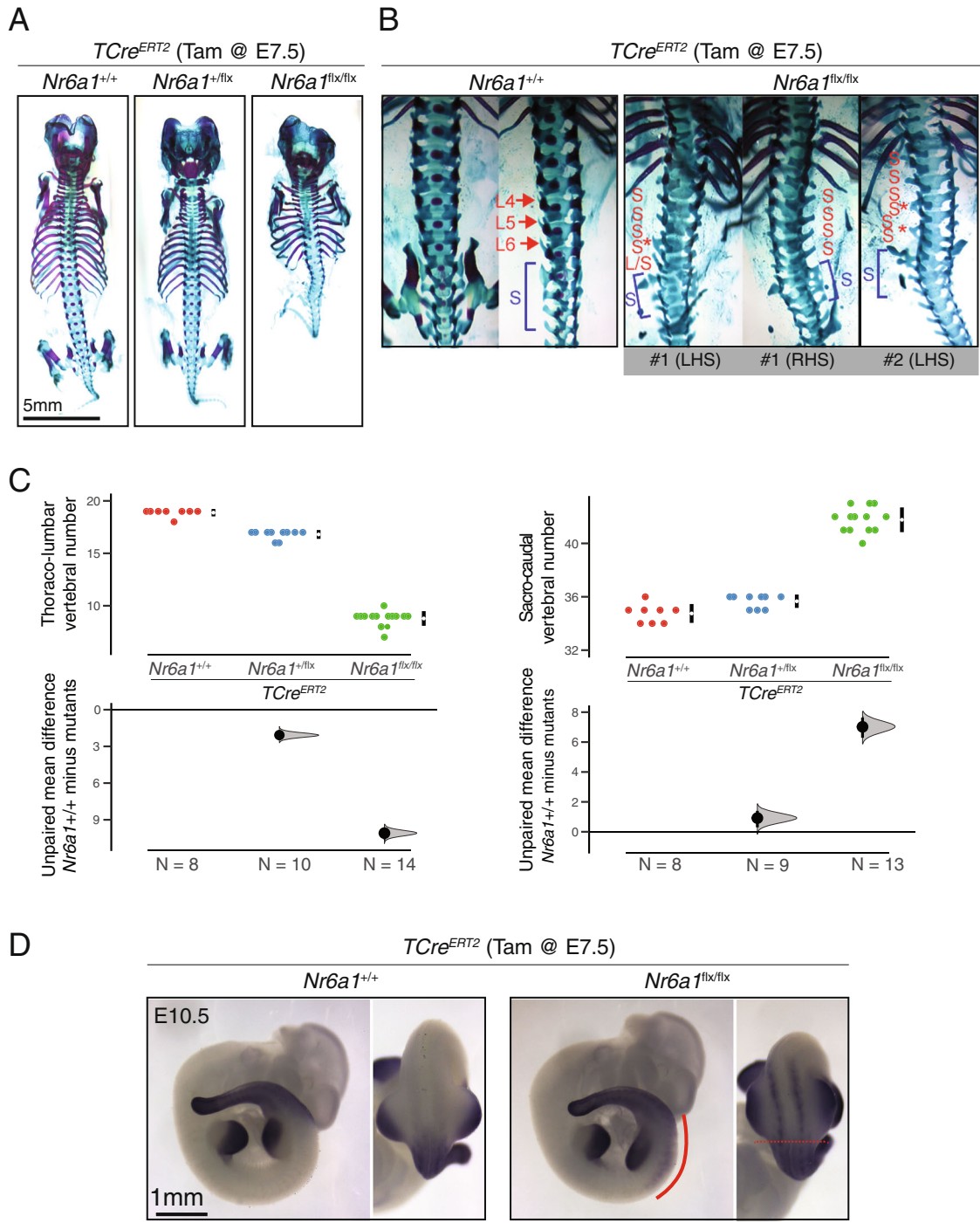

**Fig. 2 | Nr6a1 is required for trunk elongation and timely activation of sacro-caudal identity. A** Skeletal preparation of E18.5 embryos following conditional deletion of *Nr6a1* revealed a dose-dependent reduction in the number of thoraco-lumbar elements. The number of biologically independent samples assessed: *Nr6a1*[+/+] *n* = 8; *Nr6a1*[+/−] *n* = 10; *Nr6a1*[−/−] *n* = 14. **B** In WT skeletons, removal of pelvic bones allowed visualisation of transverse process morphology characteristic of lumbar (L) and sacral (blue S) elements. In two representative *TCre*[ERT2];*Nr6a1*[flx/flx] embryos, positioning of the normal sacral region is defined by the presence of rudimentary pelvic bones (blue S) and lumbar elements that have assumed sacral identity marked (red S). Red asterisk indicates lateral fusions between two adjacent elements. LHS = left-hand side; RHS = right-hand side. **C** Quantification of thoraco-lumbar and sacro-caudal vertebral number following conditional deletion of *Nr6a1*

alleles. Raw data is presented in the upper plots. Mean differences relative to wildtype (+/+) are presented in the lower plots as bootstrap sampling distributions. The mean difference for each genotype is depicted as a black dot and 95% confidence interval is indicated by the ends of the vertical error bar. **D** Whole mount in situ hybridisation for *Hoxd11* in E10.5 embryos revealed a rostral shift in the anterior limit of expression in *TCre*[ERT2];*Nr6a1*[flx/flx] embryos compared to control. Ectopic *Hoxd11* expression in *TCre*[ERT2];*Nr6a1*[flx/flx] embryos is marked with a red line in lateral view. In dorsal view, the normal anterior limit is marked with a red dotted line, with ectopic *Hoxd11* expression restricted to the paraxial mesoderm. Consistent spatial changes were observed in 3 biologically independent samples/genotype.

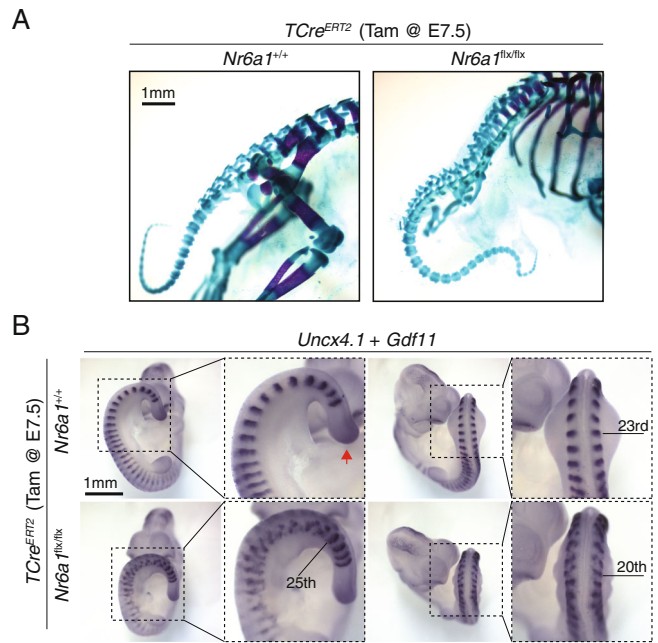

**Fig. 3 | Nr6a1 function within somitic tissue is restricted to the trunk region.**
**A** Analysis of caudal elements in wildtype and *TCre^ERT2;Nr6a1^flx/flx* and *Nr6a1^flx/flx*
embryos (*n* = 13 and *n* = 8 biologically independent samples, respectively) revealed
no difference in vertebrae morphology or number. **B** Whole mount in situ hybri-
disation for the somite marker *Uncx4.1* revealed a rostral shift in hindlimb posi-
tioning in *TCre^ERT2;Nr6a1^flx/flx* embryos compared to control, somite number is
marked in the right panel. Additionally, somite morphology was specifically dis-
rupted in the future thoraco-lumbar region of *TCre^ERT2;Nr6a1^flx/flx* embryos, returning
to normal immediately after the hindlimb bud. Co-expression analysis of *Gdf11* (red
arrows) revealed no change in expression in the tailbud between genotypes. Con-
sistent expression patterns were observed in 2 biologically independent samples/
genotype.

developmental program but rather, Nr6a1 clearance is essential for
correct sacral positioning and normal tail morphology.

To ascertain whether *Hox* expression dynamics were altered in
*Cdx2P:Nr6a1* embryos, we performed RNA-seq analysis of tailbud tis-
sue at E10.5, a time when Nr6a1 should no longer be expressed at this
site. Compared to wildtype controls, the prolonged maintenance of
Nr6a1 activity resulted in a modest upregulation of several trunk *Hox*
genes, including *Hoxc6, Hoxc8, Hoxa9, Hoxc9* and *Hoxc10*, genes
known to participate in conveying T-L identity[49,51,54] (Fig. 5C). Con-
versely, genes of the *Hox11-13* paralog groups and two 5′-Hox cluster
antisense genes *Hoxa11os* and *Hottip* showed reduced expression in
*Cdx2P:Nr6a1* tailbuds (Fig. 5C), with *Hoxb13* again having the largest
differential response to altering Nr6a1 levels. One exception to this
trend was *Hoxa11*, which was found to be upregulated in *Cdx2P:Nr6a1*
tailbuds. A similar inverse relationship between *Hoxa11* and other
posterior *Hox* cluster genes, including *Hoxa11os*, has also been
observed in E10.5 *Gdf11;miR-196*single and compound mutant mice
which harbour additional thoraco-lumbar elements[44]. An antagonistic
relationship between *Hoxa11os* and *Hoxa11* exists within the early
mouse limb bud, resolving into mutually exclusive domains of
expression as the limb bud elongates[55]. In this context, *Hoxa11os*
transcription is dependent on Hoxa13/d13 and suppresses *Hoxa11*, with
our data supporting the potential for a parallel regulatory network
within the paraxial mesoderm.

We went on to assess the expression of *Hoxd12* by whole mount
in situ hybridisation in E10.5 *Cdx2P:Nr6a1* embryos, revealing a general
reduction in expression throughout the entire tail compared to WT
somite-matched embryos, with clear reductions observed in tailbud

mesoderm and somitic tissues (Fig. 5D). In summary, we show that the
Nr6a1, likely acting with as-yet unknown protein partners, is sufficient
to control the temporal progression of *Hox* activation of all 4 *Hox*
clusters. This control over *Hox* cluster progression by Nr6a1 correlated
well with altered patterning outcomes observed following manipula-
tion of Nr6a1 levels. However, the increase in T-L vertebral number
seen in *Cdx2P:Nr6a1* embryos was harder to reconcile, since meristic
changes (i.e. changes in vertebral number) are historically not thought
to ensue downstream of changes to a single *Hox* gene or Hox paralog
group. Growing evidence however has challenged this dogma[44,50]
supporting the view that the observed altered *Hox* code may indeed
drive meristic changes.

## Opposing effects of Nr6a1 and Gdf11 on a trunk regulatory module

The similarity of T-L expansion phenotypes seen in Nr6a1 gain-of-
function (Fig. 5A) and Gdf11 loss-of-function[22] mice prompted us to
compare the global molecular changes identified in our E10.5
*Cdx2P:Nr6a1* RNAseq dataset with those of a published E10.5 *Gdf11^−/−*
tailbud dataset[28]. This analysis revealed 151 conserved differentially
expressed genes (Fig. 5D; Supplementary Data 2), 90% of which dis-
played the same direction of regulation in both mutants (Fig. 5F). 63 of
the 136 co-regulated genes were upregulated, including genes enri-
ched in NMPs (*Nkx1-2, Epha5, Gpm6a* and *Gldc*)[20], as well as genes
known to promote axial elongation such as *Lin28a, Lin28b*[27–29]. On the
other hand, 73 of the 136 co-regulated genes were downregulated and
included many posterior *Hox* genes, supporting an antagonistic
arrangement whereby Nr6a1 promotes and Gdf11 terminates a core
trunk gene regulatory network.

To test the in vivo epistatic interaction between Nr6a1 and Gdf11, we
set out to assess whether Nr6a1 mediates Gdf11-dependent phenotypes
through compound mutant analysis. Conditional deletion of Nr6a1 using
the *T-Cre^ERT2* deleter line was repeated with Tam administration at E8.5,
one day later than previous experiments and at a time when deletion was
anticipated to have little to no phenotypic consequence that would
confound compound mutant analysis. Indeed, no change in TVN was
observed in *TCre^ERT2;Nr6a1^flx/flx* embryos when Tam was administered at
E8.5, on an otherwise wildtype (*Gdf11^+/+*) or *Gdf11^+/−* background (Sup-
plementary Fig. 3A). However on a *Gdf11^−/−* background which is known
to display an expanded T-L count of 27 vertebrae (T18 L9; Fig. 5B, C and
ref. [22]), conditional deletion of Nr6a1 at E8.5 significantly rescued T-L
count by 3-4 elements (T16/17 L7), with Nr6a1 dose-dependent effects
revealed (Fig. 6A, B). This temporal deletion of Nr6a1 did not rescue
aberrant tail vertebrae morphology nor tail truncation known to occur in
*Gdf11^−/−* embryos, consistent with earlier interpretation that Nr6a1
function was not required at this site. This work identifies separable
phenotypes, trunk elongation and tail morphology, that each depend on
Gdf11 activity but utilise different downstream mechanisms. Collec-
tively, this work establishes Nr6a1 as a master intrinsic regulator of T-L
vertebral number and morphology, and reveals that termination of
primary body axis formation by Gdf11 is mediated, likely to a large
extent, through the clearance of *Nr6a1* expression.

## Nr6a1, along with the external factors, regulates *Hox* timing

To more precisely delineate the upstream signals influencing *Nr6a1*
expression, and the downstream molecular events requiring Nr6a1
activity, we turned to an in vitro embryonic stem cell (ESC) differ-
entiation approach that models the developmental kinetics of a pos-
terior growth zone by the sequential addition of FGF2 on day (d) 0,
WNT pathway agonist CHIR99021 on d2, with or without the addition
of Gdf11 on d3 (Supplementary Fig. 8A; based on refs. [20,56]). The loss
of epiblast markers (Supplementary Fig. 8B), the induction of NMP
identity (Supplementary Fig. 8B, C) and 3′ *Hox* cluster activation
(Supplementary Fig. 8D) were all confirmed soon after Wnt activation.
Collinear trunk *Hox* activation ensued quickly thereafter, with

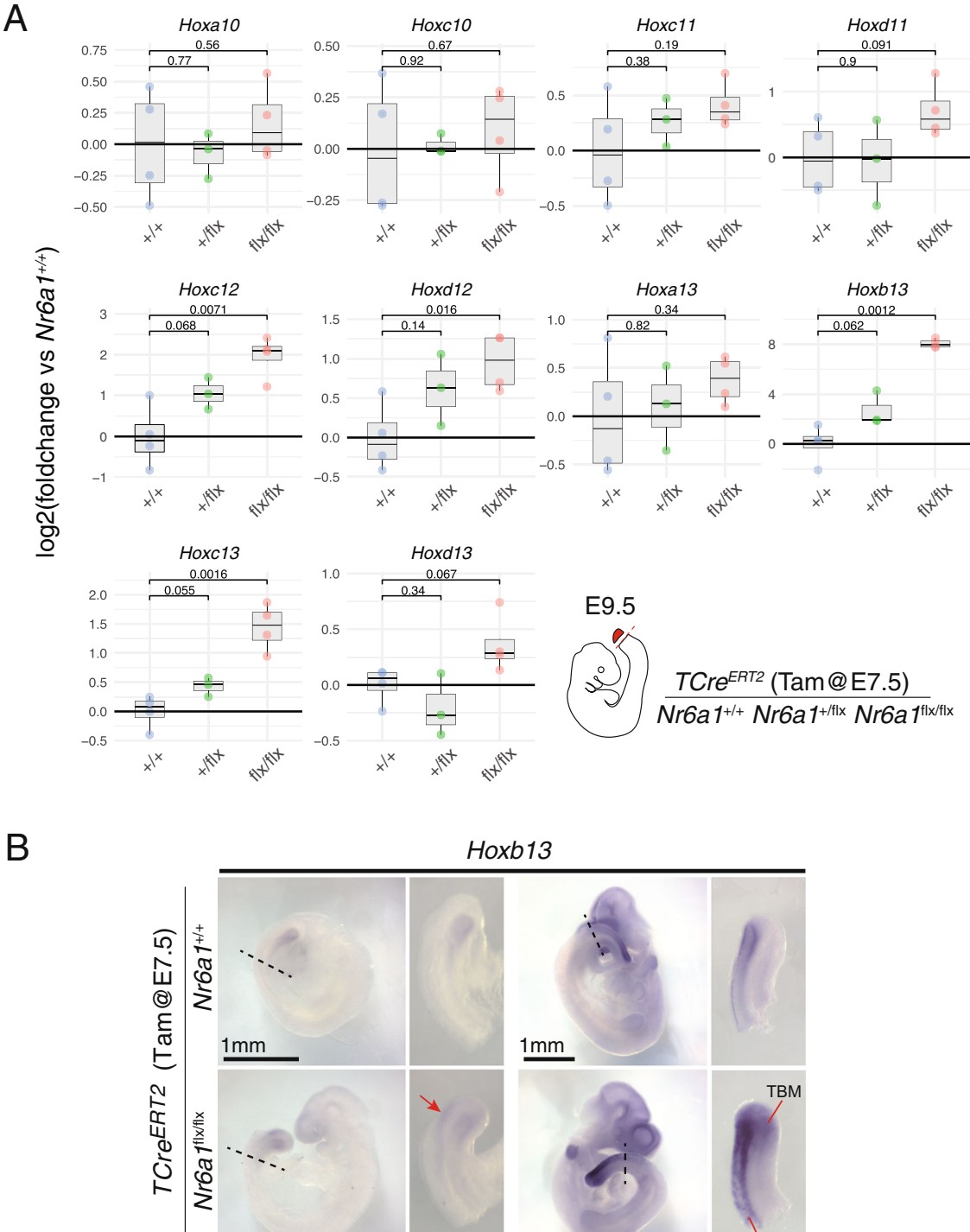

**Fig. 4 | Nr6a1 suppresses posterior *Hox* gene expression. A** Quantitative PCR analysis of posterior *Hox* gene expression within tailbud tissue isolated from *TCre^ERT2* controls, *TCre^ERT2;Nr6a1^{+/flx}* and *TCre^ERT2;Nr6a1^{flx/flx}* embryos at E9.5, *n* = 3 biologically independent samples per genotype. Statistical analysis of differences between genotypes was performed using an unpaired two-sample Welch's *t* test, *p* values are provided in the figure. Source data are provided as a Source Data file. **B** Whole mount in situ hybridisation revealed a spatial expansion of *Hoxb13* expression in *TCre^ERT2;Nr6a1^{+/flx}* embryos compared to wildtype, at both E9.5 and E10.5. Red arrow at E9.5 indicates ectopic *Hoxb13* expression. TBM = tail bud mesoderm; NT = neural tube.

progression through to a posterior *Hox* code (Supplementary Fig. 8D) and a robust sacro-caudal producing E9.5-NMP molecular signature (Supplementary Fig. 8E) requiring Gdf11.

In wildtype cells, *Nr6a1* was present in epiblast-like d2 cells and its expression was substantially enhanced on d2.5 and d3 following the

induction of an NMP-like state (Supplementary Fig. 9A). While *Nr6a1* expression was already decreasing after d3, it was sharply terminated following the addition of exogenous Gdf11 (Supplementary Fig. 9A), consistent with earlier in vivo results. Interestingly, an inverse temporal pattern of expression was observed for *Nr6a1os*, remaining very

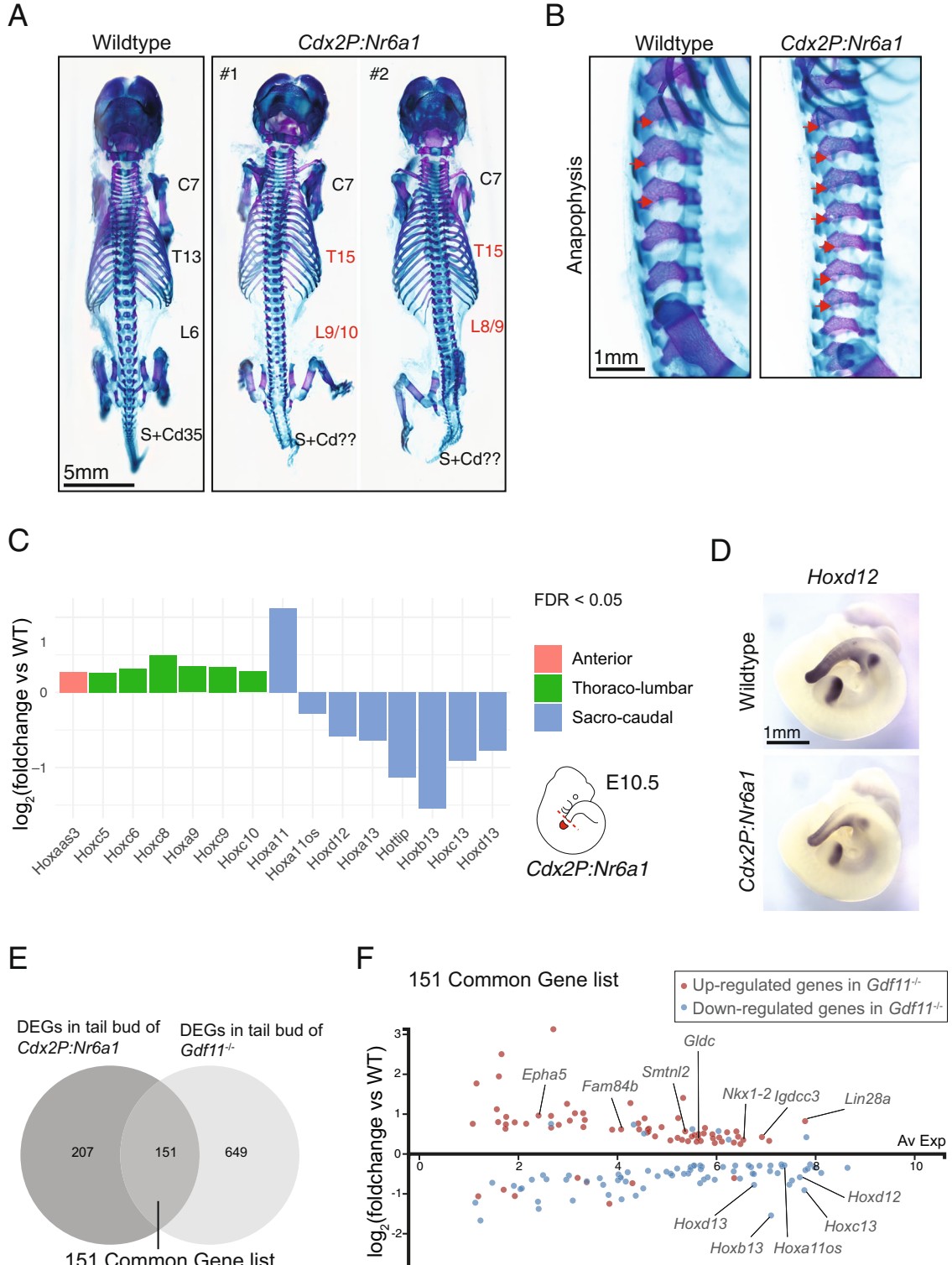

low until the addition of Gdf11 when levels increased 4-fold (Supplementary Fig. 9A). We find that the promoter of *Nr6a1os* is accessible in epiblast stem cells treated with or without Chiron for 48 h (Supplementary Fig. 9B)[57], and harbours several consensus binding motifs for Smad2/3 (Supplementary Fig. 9C), ultimate transducers of Gdf11 signalling[24] and Smad4, the common-mediator Smad, which forms heterotrimeric complex with Smad2/3 to control the target genes[58]. This raises the intriguing possibility that Gdf11 suppression of *Nr6a1* sense transcripts is in fact mediated via a direct activation of *Nr6a1os* and subsequent genomic interactions between sense and antisense loci that establish mutually exclusive expression patterns, as has been observed for other developmental loci[55].

With wildtype differentiation conditions established, we next generated *Nr6a1*[-/-] ESC clones by CRISPR/Cas9[59,60], deleting the DNA-binding domain from both alleles (Supplementary Fig. 10). These *Nr6a1*[-/-] ESCs were able to generate NMPs with equal kinetics to wildtype cells (Supplementary Fig. 11A–D), though whether the temporal progression of axial identity within these NMPs was altered, as may be predicted from the in vivo deletion of Nr6a1, remained unclear.

**Fig. 5 | Prolonged Nr6a1 activity is sufficient to increase thoraco-lumbar count and sustain trunk expression signatures. A** Skeletal preparation of E18.5 embryos, dorsal view, revealed an increase in the number of thoraco-lumbar elements in *Cdx2P:Nr6a1* embryos (*n* = 2 independent samples) compared to wildtype (*n* = 8 independent samples). C = cervical, T = thoracic, L = lumbar, S = sacral and Cd = caudal vertebrae. **B** Higher magnification, lateral view, of the lumbar region in embryos from (**A**) revealed the anapophysis (red arrow) normally present on the first 3 lumbar elements in wildtype was now present on 7 of the 8 lumbar elements in *Cdx2P:Nr6a1* embryos. Experimental numbers as per 5A. **C** RNAseq analysis of E10.5 wildtype and *Cdx2P:Nr6a1* tailbuds revealed widespread changes in *Hox* expression downstream of Nr6a1. Results are presented as a log2-transformed fold change in *Cdx2P:Nr6a1* samples relative to wildtype, *n* = 2 for *Cdx2P:Nr6a1* and *n* = 4 for wildtype. Only those *Hox* genes with a false discovery rate (FDR) < 0.05 are displayed, and are colour-coded based on the axial region where the Hox protein functions. Source data are provided as a Source Data file. **D** Whole mount in situ hybridisation revealed reduced expression of *Hoxd12* within the tail of E10.5 *Cdx2P:Nr6a1* embryos compared to wildtype. Consistent spatial changes were observed in 2 biologically independent samples/genotype. **E** Venn diagram overlay of differentially expressed genes (DEGs) in *Cdx2P-Nr6a1* (this work) and *Gdf11⁻/⁻* [28] tailbud tissue reveals substantial overlap. Source data are provided as a Source Data file. **F** Analysis of the 151 co-regulated genes identified in (**D**), presented here as log2-transformed fold change in *Cdx2P:Nr6a1* samples relative to wildtype, revealed that 136 genes (90%) displayed the same direction of regulation in both genetically altered backgrounds. Red and blue dots indicate upregulated and downregulated genes in *Gdf11⁻/⁻* tailbuds, respectively. Source data are provided as a Source Data file.

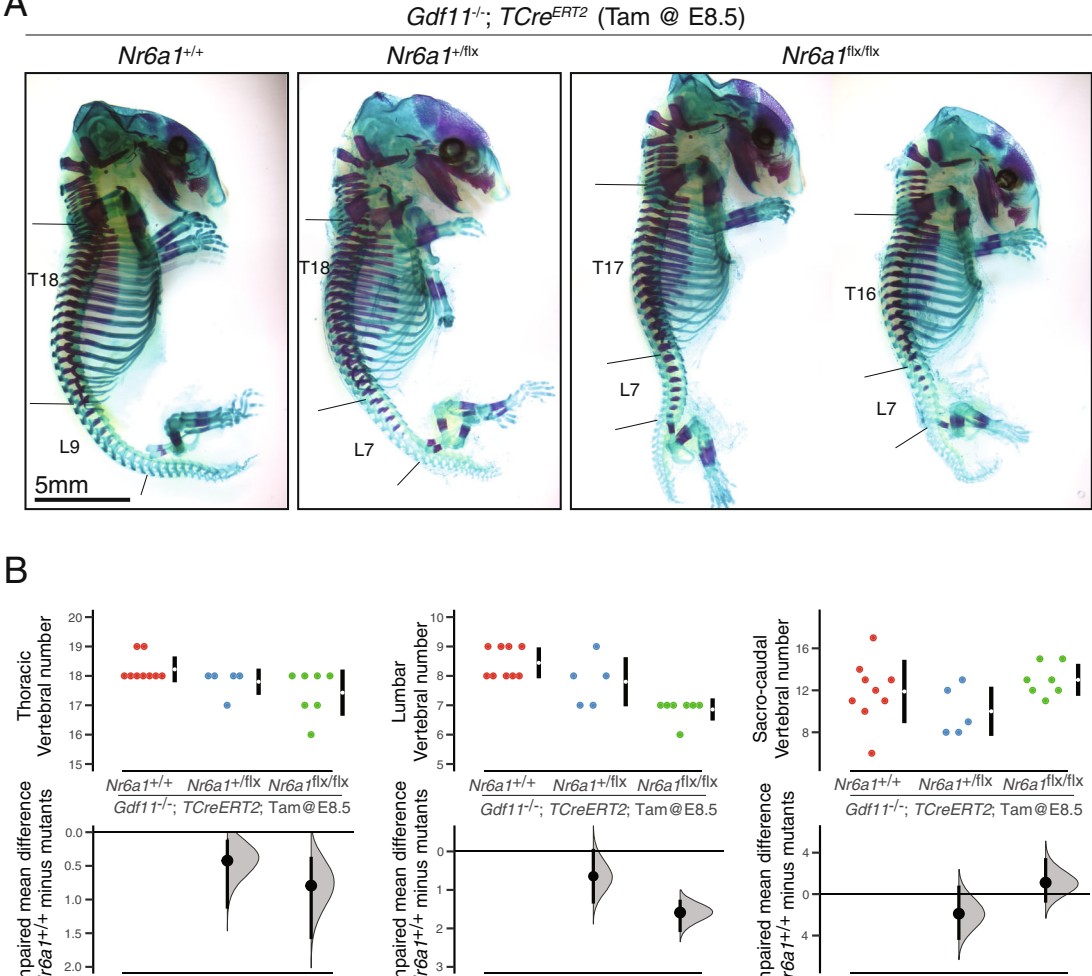

**Fig. 6 | Gdf11 signalling terminates Nr6a1-dependent trunk elongation.**
**A** Skeletal preparation of E18.5 embryos, lateral view, revealed a dose-dependent rescue of *Gdf11⁻/⁻* thoraco-lumbar expansion following conditional deletion of *Nr6a1* alleles. The number of biologically independent samples assessed: *Nr6a1⁺/⁺;Gdf11⁻/⁻ n* = 9; *Nr6a1⁺/⁻;Gdf11⁻/⁻ n* = 5; *Nr6a1⁻/⁻;Gdf11⁻/⁻ n* = 7. T = thoracic, L = lumbar. **B** Quantification of thoracic, lumbar and sacro-caudal vertebral number across genotypes. Raw data is presented in the upper plots. Mean differences were calculated relative to *Gdf11⁻/⁻;Nr6a1⁺/⁺* and presented in the lower plots as bootstrap sampling distributions. The mean difference for each genotype is depicted as a black dot and 95% confidence interval is indicated by the ends of the vertical error bar.

To this end, we performed RNAseq analysis as differentiation proceeded, and first assessed expression of the known Nr6a1 target[43] and E8.5-NMP marker[20], *Pou5f1* (*Oct4*). *Pou5f1* expression was slightly increased on d2 before the addition of Wnt, with a peak in enhanced differential expression observed on d3 (Supplementary Fig. 12B). This was accompanied by a statistical enrichment of many E8.5-NMP signature genes on d3, and concomitant reduction in E9.5-NMP signature genes (Supplementary Fig. 12C). This surprising result suggests that the deletion of Nr6a1 does not autonomously cause a speeding up of developmental timing within NMPs, and that additional factor(s) may be required for the earlier observations in vivo.

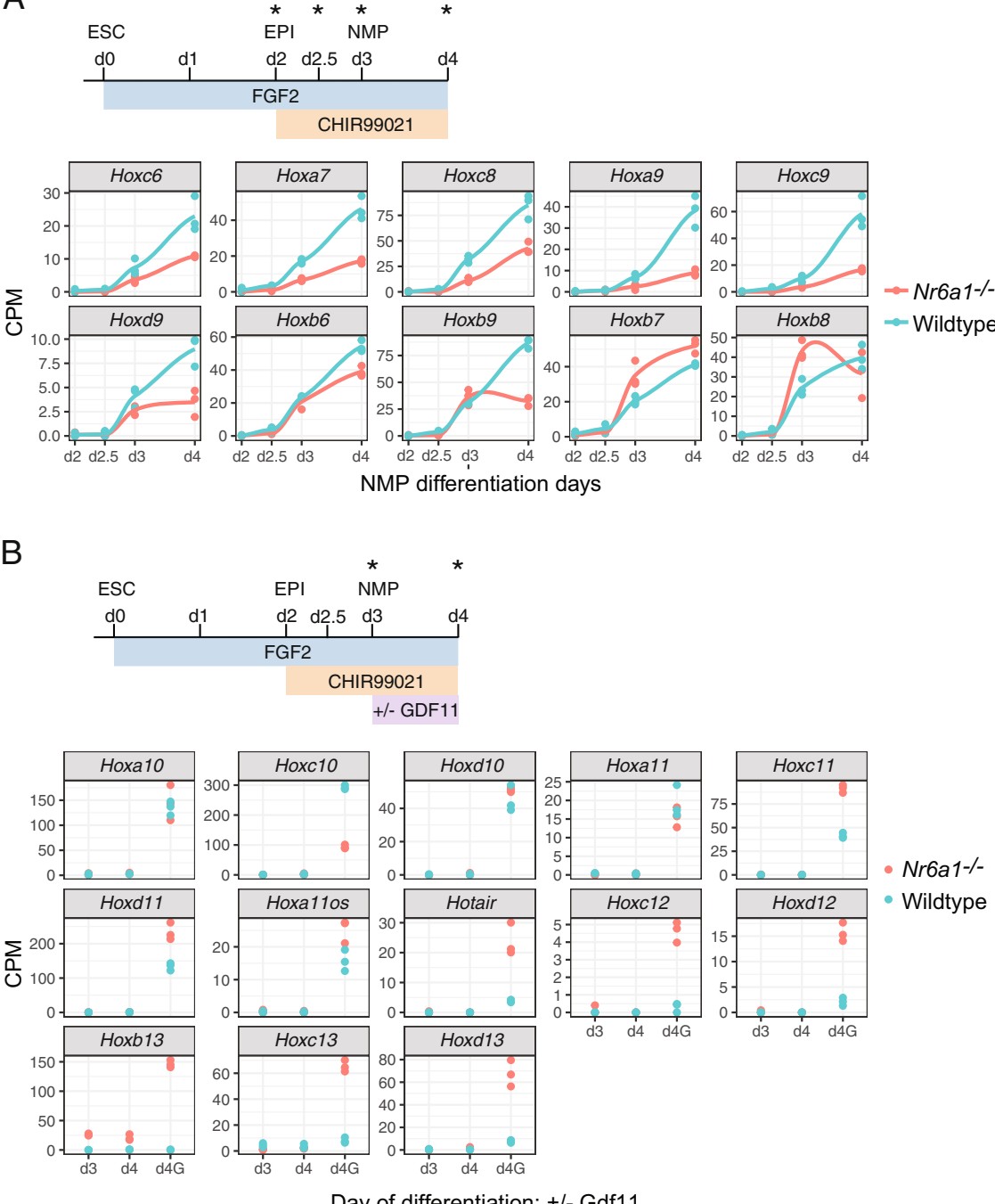

**Fig. 7 | *Nr6a1* and Gdf11 signalling cooperate in the control of *Hox* cluster transitions. A**, **B** Differential *Hox* expression kinetics observed during in vitro differentiation of *Nr6a1* and wildtype ESCs. Source data are provided as a Source Data file. **A** RNAseq analysis of trunk *Hox* gene expression (*Hox6-9*) in *Nr6a1*[-/-] (red) and wildtype (blue) cells. In vitro differentiation protocol schematised, analysis timepoints marked with asterisk and data represented as counts per million (CPM).

Only those genes showing a false discovery rate (FDR) < 0.05 are visualised. **B** RNAseq analysis of posterior *Hox* gene expression (*Hox10-13*) in *Nr6a1*[-/-] (red) and wildtype (blue) cells. In vitro differentiation protocol schematised, analysis timepoints marked with asterisk and data represented as counts per million (CPM). Only those genes showing a false discovery rate (FDR) < 0.05 are visualised. G = addition of Gdf11.

We next went on to characterise *Hox* expression dynamics within this RNAseq dataset. The robust initiation of a trunk Hox code normally seen 24 h after Wnt activation was reduced in *Nr6a1*[-/-] cells particularly for genes of the *Hoxa, Hoxc* and *Hoxd* clusters, with the expression of *Hoxc6, Hoxa7, Hoxc8, Hoxa9, Hoxc9* and *Hoxd9* all diminished relative to WT on d3 (Fig. 7A). This relative reduction was exacerbated even further when cells were cultured in Fgf/Wnt alone

for a further 24 h. Genes of the *Hoxb* cluster showed mixed responses in *Nr6a1*[-/-] cells, however collectively, these observed expression signatures support the view that Nr6a1 does not serve a major role in the initiation of trunk Hox gene expression but is required to achieve their full expression potential.

Under these standard Fgf/Wnt culture conditions, the activation of posterior *Hox10-13* genes was almost universally absent in cells of

both genotypes (Fig. 7B). For wildtype cells this is expected, since Gdf11 is required in these cultures for robust activation (Supplementary Fig. 8D)[61,62]. For *Nr6a1*[-/-] cells, however, these results allowed us to build a molecular hierarchy which was not able to be deciphered from in vivo experiments: the loss of Nr6a1 in general is not sufficient on its own to activate posterior *Hox* expression and requires positive input from additional signal(s). Once Gdf11 was exogenously applied to these cultures, *Nr6a1*[-/-] cells activated posterior *Hox* expression with far greater amplitude than wildtype cells, with this heightened response to Gdf11 more evident the further 5′ within a cluster the gene was located (Fig. 7B and Supplementary Fig. 12D).

One striking exception to this identified hierarchy was *Hoxb13*. 24 h after the activation of Wnt, and without any exogenous Gdf11, *Hoxb13* expression was induced 25-fold relative to wildtype cells where a complete absence of expression was expected and observed (Fig. 7B). Expression persisted at this level without further signal input, and upon addition of Gdf11, *Hoxb13* showed the largest differential increase in expression relative to wildtype of any posterior *Hox* gene (150-fold). The mechanism behind this unique regulation of *Hoxb13* versus other posterior *Hox* genes remains to be defined, though is consistent with the heightened regulation of *Hoxb13* in Nr6a1 in vivo gain-of-function experiments. With this exception noted, our in vitro results globally reinforce the dominant role of Gdf11 in activating posterior *Hox* expression, and additionally builds in an essential function for Nr6a1 in suppressing the untimely/precocious activation of posterior *Hox* expression and in defining their precise levels once expression has been activated.

### *Nr6a1* biases cell lineage choice in vitro and in vivo

The allocation of NMP descendants towards neural or mesodermal lineages is guided by both extrinsic[63–65] and intrinsic factors[66,67], with a disproportionate allocation to one lineage often resulting in termination of elongation[68,69]. Progenitor cells within the *Gdf11*[-/-] tailbud are biased toward a neural fate and an enlarged neural tube forms within caudal regions of these truncated mutant embryos[28]. In other developmental contexts, Nr6a1 activity has been linked to neural specification and differentiation in vitro[70,71] and in vivo during mouse[42] and *Xenopus*[72,73] development, raising the possibility that Nr6a1 may promote neural cell fate within bipotent NMPs.

To address this, we first analysed our various RNAseq datasets for differential expression (FDR < 0.05) of genes previously found to delineate mesodermal progenitors (MP) or neural progenitors (NP) of the E8.5 tailbud[20]. Interrogation of in vitro differentiation samples 24 h after Wnt activation revealed heightened expression of MP genes and reduced expression of NP genes within *Nr6a1*[-/-] cells compared to wildtype (Fig. 7A; Note: *Aldh1a2* in this context is a marker of MPs[20]). Conversely, interrogation of in vivo *Cdx2P:Nr6a1* tail buds revealed the opposite, NP genes were found to be upregulated while MP genes were downregulated when Nr6a1 was ectopically maintained (Fig. 7B). While some of these lineage markers assessed are not uniquely restricted to these lineages, particularly at differentiation stages, the inclusion of key lineage markers within tailbud progenitor stages, such as *Sox1* (NP) and *Tbx6* or *Msg1* (MP), supported further investigation of a potential bias in lineage choice in vivo.

To this end, we performed Flow cytometry analysis of E9.0 wildtype and *TCre*[ERT2]*;Nr6a1*[flx/flx] tailbuds, to quantify any shift in the numbers of T/Bra⁺; Sox2⁺ (NMPs), T/Bra^neg; Sox2⁺ (NP) and T/Bra⁺; Sox2^neg (MPs). The time point is important, since it is when cells that will go on to form the thoraco-lumbar region - the axial region that is strongly dependent on Nr6a1 function - are transiting through the PSM. While we observed no difference in the number of NMPs between WT and *TCre*[ERT2]*;Nr6a1*[flx/flx] tailbuds we saw a significant decrease in NPs and increase in MPs following loss of Nr6a1 (Fig. 8C; WT n = 4 and CKO n = 9 individual tailbuds across two litters), consistent with the molecular biases observed (Fig. 8A, B). Together, these results identify *Nr6a1* as

an intrinsic factor that has a minor role in regulating the balance between neural and mesodermal fates within NMPs during axial elongation.

## Discussion

The initial stages of vertebral column formation require many cellular decisions and tissue-level processes that, despite their required coordination, are often considered separately in terms of genetic regulation. Here, we have identified a single factor that coordinately controls vertebral number and identity, somite segmentation and NMP cell lineage choice in the mouse (summarised in Fig. 9). The axially-restricted manner in which Nr6a1 functions reinforces the existence of distinct developmental programs controlling trunk vs. tail formation, emphasising that this is not a developmental continuum that is delineated by a switch in patterning program and progenitor location alone. Rather, this switch involves additional changes in the genetic regulation of segmentation and regional control of vertebral number, all processes unified in their requirement for precise levels of Nr6a1 activity.

Complete loss of Nr6a1 in the mouse was found to slow the overall pace of development soon after the initiation of somitogenesis and, morphologically, these null embryos failed to progress through the period of embryonic turning (E8.75)[30]. Nonetheless, up until approximately E9.5 when *Nr6a1*[-/-] embryos died, the very anterior and posterior embryonic ends continued to expand and differentiate while the trunk region stalled, consistent with our collective analyses establishing Nr6a1 as a selective regulator of trunk development. *Nr6a1* transcripts have been detected ubiquitously in the mouse embryo at E6.5[30,43], however, the first 7-13 somites still form in the complete absence of Nr6a1 activity[30], broadly correlating with the future cervico-thoracic boundary. Here, *Nr6a1* expression levels rise sharply in cells of the posterior embryo central to axial elongation[41], a site high in Wnt activity[74] which we show in vitro has the ability to rapidly enhance *Nr6a1* transcript levels. As the T-L region is being laid down, Nr6a1 levels remain high, and a correlation between caudal *Nr6a1* expression level and total T-L vertebral number was observed across our *miR-196-TKO;Gdf11*[-/-] deletion series. Indeed, *Nr6a1* levels were shown to be quantitatively instructive based on the dose-dependent phenotypes observed in both Nr6a1 gain- and loss-of-function scenarios. As development proceeds and the primary body elongation program comes to its eventual completion, we show an essential requirement for the clearance of *Nr6a1* expression within the tailbud. This clearance was achieved by multiple mechanisms, including the rising levels of Gdf11 signalling as well as microRNA repression by the posteriorly-expressed miR-196 paralogs.

The 3′UTR of *Nr6a1* in fact houses binding sites for several microRNAs, including another Hox-embedded miRNA family, *miR-10*, as well as the developmental timing microRNA family, *let-7* (Supplementary Fig. 1C). *miR-10* paralogs are genomically positioned between *Hox4/5* paralogs and thus are expressed from early stages of axial elongation[75,76]. This raises the intriguing possibility that miR-10 may act to buffer the functional domain of Nr6a1 activity at more anterior locations, while miR-196 reinforces its posterior boundary, collectively delimiting Nr6a1 output to the T-L region in synchrony with the temporal progression of Hox cluster activation. Finally, within the tail-forming region, the rising levels of Gdf11, and decreasing levels of Nr6a1 activates expression of progressively more posterior *Hox* genes, including the Hox13 paralogs which shift the Lin28-*let-7* axis in favour of *let-7* expression[28], a potent repressor of Nr6a1 at least in vitro[39] thus reinforcing termination of expression. Collectively, this work has revealed a highly integrated series of regulatory mechanisms defining axially-restricted expression of *Nr6a1*, and ultimately, defining vertebral number within the trunk region of the mouse.

The consequences of sustaining Nr6a1 activity in vivo for longer than normal bared resemblance to those observed following sustained

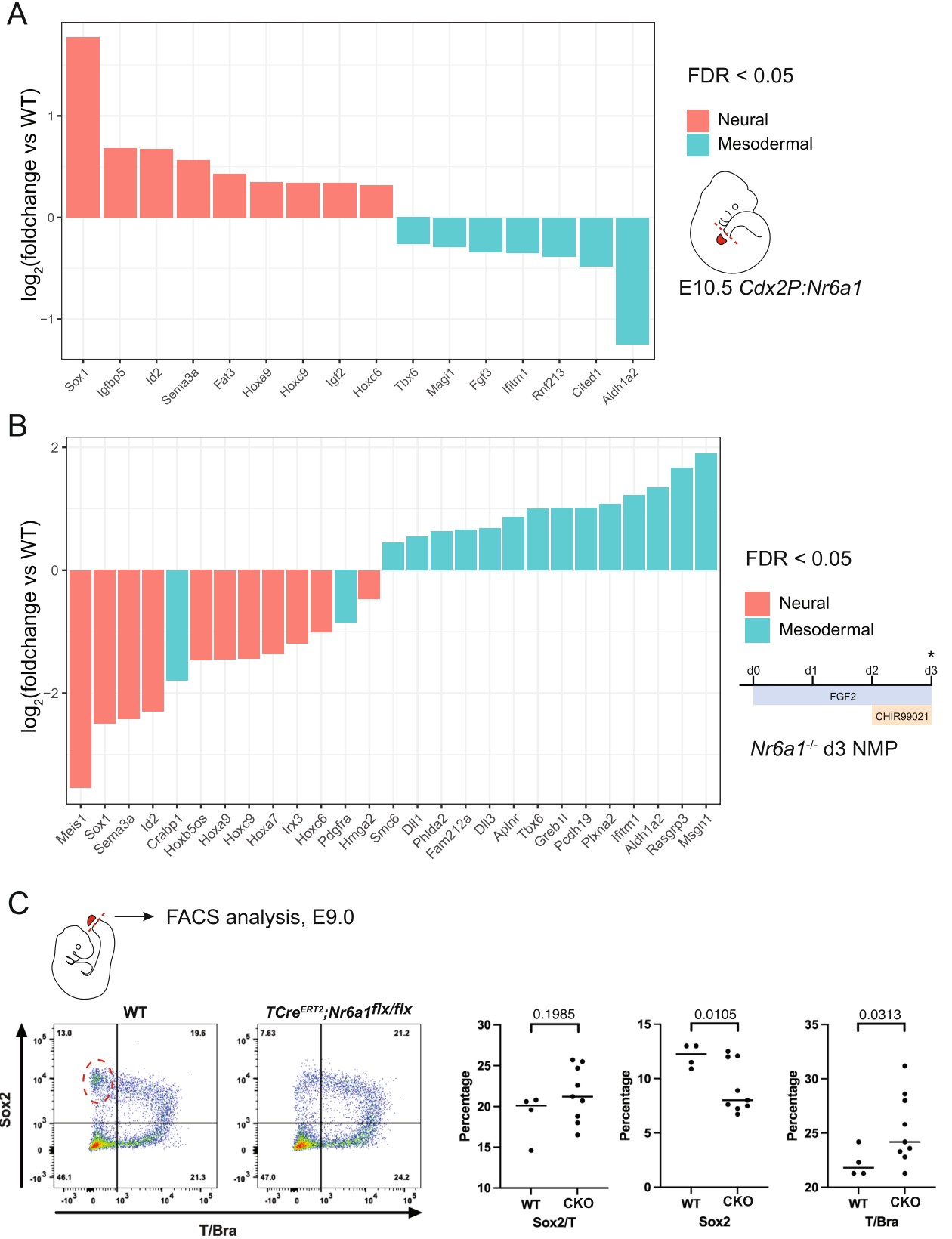

activity of the Pou-domain transcription factor Oct4/Pou5f1, both in terms of expanded trunk vertebral number and in the delayed activation of posterior *Hox* expression[26]. This parallel is counterintuitive since Nr6a1 is a well-characterised direct repressor of *Oct4* in ESCs[71,77–79] and, consistent with this view, *Oct4* expression is enhanced in the posterior growth zone of Nr6a1 null embryo at E8.5[43] and in our

equivalent-stage *Nr6a1⁻/⁻* ESC-derived NMPs (Day2.5 and Day3 NMPs; Supplementary Fig. 12B). Why then does in vivo deletion of Nr6a1 reduce trunk vertebral number if *Oct4* expression is heightened? We suggest the explanation for this result lies in the timing of gene regulatory mechanisms. While loss of Nr6a1 upregulates *Oct4* within trunk-forming progenitors (E8.5 in vivo, Day2.5/3 in vitro), it is known

**Fig. 8 | Nr6a1 activity regulates neural vs mesodermal fate choice in vitro and in vivo. A** Heightened neural (red) and reduced mesodermal (blue) gene expression signatures were observed following prolonged maintenance of Nr6a1 within the mouse tailbud. RNAseq analysis was presented as a log2-transformed fold change in *Cdx2P:Nr6a1* samples relative to wildtype, $n = 2$ for *Cdx2P:Nr6a1* and $n = 4$ for wildtype. Neural and mesodermal gene lists were selected based on in vivo single-cell RNAseq analysis[20], and only genes with a false discovery rate (FDR) < 0.05 are displayed. Source data are provided as a Source Data file. **B** Heightened mesodermal (blue) and reduced neural (red) gene expression signatures were observed in *Nr6a1⁻/⁻* in vitro-derived NMPs compared to wildtype. In vitro differentiation protocol schematised, analysis time-point (d3) marked with asterisk.

RNAseq analysis is presented as a log2-transformed fold change in *Nr6a1⁻/⁻* samples relative to wildtype, $n = 3$/genotype. Neural and mesodermal gene lists as per above, and only genes with a false discovery rate (FDR) < 0.05 are displayed. Source data are provided as a Source Data file. **C** Flow cytometry analysis of Sox2 and T/Bra protein expression within cells of the E9.0 tailbud, collected from WT ($n = 4$ biologically independent samples) and *TCreᴱᴿᵀ²;Nr6a1ᶠˡˣ/ᶠˡˣ* ($n = 9$ biologically independent samples) embryos. Single Sox2-positive, single T/Bra-positive, or dual positive cells are presented as a percentage of all single, viable cells. Statistical comparison between genotypes was performed using an unpaired t-test with Welch's correction, *p*-values are provided in the figure.

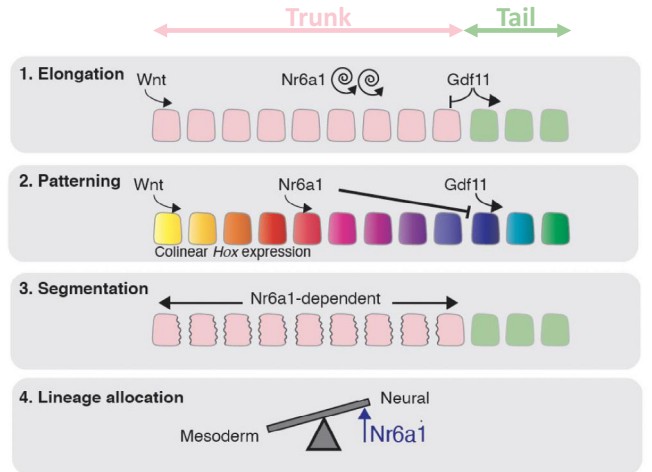

**Fig. 9 | Schematic summary highlighting the multiple roles of Nr6a1 within the tailbud.** Blocks represent somites, pink = trunk-forming somites and green = tail-forming somites.

that ectopic Oct4 activity has little to no phenotypic consequence within its normal (cervical-thoracic) domain of activity[26]. It is only at later stages, when endogenous *Oct4* is normally extinguished, that persistent/exogenous Oct4 acts to drive trunk elongation[26]. Extending our in vivo analysis to these later stages showed that *Oct4* transcripts were undetectable in both wildtype and *TCreERᵀ²;Nr6a1ᶠˡˣ/ᶠˡˣ* tailbuds at E9.5 by qPCR. Similarly, in vitro, *Oct4* expression dropped to negligible levels in both wildtype and *Nr6a1⁻/⁻* cells on d4 of differentiation (Supplementary Fig. 12D), indicating that the upregulation of *Oct4* following loss of Nr6a1 is transient. *Gdf11* and its signalling pathway components became robustly expressed with or without Nr6a1 (In vitro: d4 and d4G, Supplementary Fig. 12D; In vivo: *TCreᴱᴿᵀ²;Nr6a1ᶠˡˣ/ᶠˡˣ*, Fig. 3B), and thus would be expected to clear ectopic *Oct4* expression even in Nr6a1 loss-of-function settings, while at the same time as amplifying precocious posterior *Hox* gene expression, collectively reinforcing the premature termination of trunk formation.

In parallel with, or likely integrated with, its role in controlling axial elongation and vertebral number, this work has identified Nr6a1 as a critical intrinsic regulator of *Hox* cluster temporal progression and body patterning. Both loss- and gain-of function functional analyses supported this conclusion, with the respective precocious or delayed posterior *Hox* activation being the primary phenotypic driver in our view. For example, in *Cdx2P:Nr6a1* gain-of-function tailbuds (Fig. 5C), we do not believe the minor increase in trunk *Hox* expression seen in is the cause of the expanded thoraco-lumbar region, but rather, it is the delay in overall trunk-to-tail *Hox* code transition. Simply maintaining a trunk *Hox* code, in the absence of posterior Hox activation, would suffice. Moreover, it is possible that the minor increase in trunk *Hox* expression may be a secondary effect. In WT tailbuds, the timely activation of a posterior *Hox* code may feedback and reduce trunk *Hox*

expression levels, a scenario that is not occurring at the same time in *Cdx2P:Nr6a1* tailbuds.

At a mechanistic level, Nr6a1-dependent regulation of *Hox* cluster expression could be indirect via secondary signal(s), particularly as all 4 clusters are concurrently affected. However, our reanalysis of chromatin immunoprecipitation studies characterising mouse mesenchymal stem cells in vitro[39] has identified putative Nr6a1 binding sites within each of the four *Hox* clusters, raising the possibility of a more direct mode of regulation. To date, Nr6a1 has been characterised as a transcriptional repressor[80,81] which, in the context of the *Oct4* promoter in ESCs, recruits the methyl transferases such as Dnmt3a, Mdb2 and Mdb3 to silence expression[78,79]. Additionally, Nr6a1 has been shown in vitro to interact with the histone deacetylase complex component Nuclear receptor corepressor 1 (Ncor1)[80] and Ubiquitin interacting motif containing 1 (UIMC1)[82] in the repression of downstream target genes. Regarding the latter, the particular SNP identified within Nr6a1 as associated with additional trunk vertebrae in pigs was shown to enhance binding between Nr6a1 and Ncor1/UIMC1[35], supporting the potential for a direct mode of Nr6a1 regulation in vivo. While mechanism remains to be elucidated, the global effect of Nr6a1 in regulating the temporal activation of expression from all *Hox* clusters supports an ancient role for this nuclear receptor, at least back to the common ancestor of vertebrates when a single *Hox* cluster existed.

An unexpected aspect of Nr6a1 function revealed in these studies was its axially-restricted role in segmentation. Thoraco-lumbar segmentation required the presence of Nr6a1, though was resistant to exaggerated Nr6a1 levels. Conversely, the active clearance of Nr6a1 function was essential in allowing normal post-sacral segmentation. This switch in the requirement for Nr6a1 activity in either scenario was incredibly sharp, visualised both in terms of somite and vertebrae morphology, supporting a very rapid transition in the genetic networks driving this developmental process.

Axially-restricted contribution to pre-sacral or post-sacral segmentation has been observed in a small number of studies, highlighting potential downstream effectors of Nr6a1 in this context. For example, some *Hox* genes have been shown to cycle in the PSM[83] and, while altered *Hox* expression or Hox function has not generally not been linked to defects in segmentation, the in vivo ectopic expression of a linker region-deficient Hoxb6 protein was shown to cause severe defects in segmentation posterior to the hindlimb only[84]. In the case of Nr6a1 deficiency, Hox protein function is not expected to be altered, though it remains possible that aberrant levels of Hox activity may impact segmentation. A second, and more likely, mediator is Lunatic Fringe (Lfng), a core component/inhibitor of the Notch signalling pathway. Throughout segmentation, *Lfng* is expressed both in an oscillatory pattern in the posterior PSM and stably at the anterior PSM border[85,86], controlled by independent enhancers[87,88]. The specific abolition of oscillatory *Lfng* expression via enhancer deletion resulted in highly malformed vertebrae of reduced number specifically within the presacral region, with the tail extending largely unaffected[89–91]. This regional phenotype was very similar to what was observed in *TCre:Nr6a1⁻/⁻* mutant embryos (Fig. 2F), and moreover, cyclic expression of *Lfng* has been found to be greatly reduced in *Nr6a1⁻/⁻* embryos

while the stable band of *Lfng* in anterior PSM was unaffected[30]. Collectively, this supports the oscillatory expression of *Lfng*, downstream of Nr6a1, as essential for presacral somite segmentation.

While we have focused on the paraxial mesoderm and its major derivative, it is clear that the entire trunk region expands or contracts following manipulation of Nr6a1 level, suggesting that cell-autonomous Nr6a1 actions are likely to extend across multiple tissue/germ layers in addition to more global mechanisms of tissue-level coordination. For the lateral plate mesoderm, a change in hindlimb positioning that mirrors the anterior or posterior displacement of sacral vertebrae has been observed in Gdf11 signalling gain- and loss-of-function mouse mutants[22,25] and also here in *Cdx2P:Nr6a1* gain-of-function embryos. In *TCre:Nr6a1⁻ᐟ⁻* embryos however, the positioning of rudimentary pelvic bones maintained an almost "wildtype" distance from the last formed rib-bearing thoracic element, despite all intervening elements being transformed from lumbar to sacral identity. This suggests a likely dissociation in Nr6a1's regulation of *Hox* signatures between tissue layers, and in the downstream events they instruct[92]. From an evolutionary perspective, our work provided experimental support for changes in Nr6a1 activity underlying the modest, albeit economically-beneficial, changes in axial formulae observed in domesticated pigs and sheep. Whether analogous changes are widespread across the vertebrates and whether more dramatic changes in *Nr6a1* expression or function may have supported the evolution of extreme phenotypes such as the elongated vertebral column in snakes remain important areas for future investigation.

## Methods

### Experimental models and subject details

**Animal models and ethical approval.** All animal procedures in Monash University were performed in accordance with the Australian Code of Practice for the Care and Use of Animals for Scientific Purposes (2013). These experiments were approved by the Monash Animal Ethics Committee under project number 21616. Animal experiments performed at the Stowers Institute for Medical Research were conducted in accordance with an Institutional Animal Care and Use Committee approved protocol (IACUC #2019-097).

### Mice

*Gdf11⁻ᐟ⁻* [22], *Nr6a1⁻ᐟ⁻* [30], *Nr6a1ᶠˡˣ/ᶠˡˣ* [46], *CMVCre* [47] and *TCreᴱᴿᵀ²* [48] mouse lines have been previously described, and were maintained on a C56BL/6 background. *TCreERᵀ²* and *Nr6a1ᶠˡˣ/ᶠˡˣ* lines were intercrossed to generate *TCreᴱᴿᵀ²;Nr6a1⁺/ᶠˡˣ* males, which were subsequently bred with *Nr6a1⁺/ᶠˡˣ* or *Nr6a1ᶠˡˣ/ᶠˡˣ* dams for time-mate collection of embryos. A similar breeding strategy was employed to incorporate the *Gdf11⁺/⁻* allele onto this *TCreᴱᴿᵀ²;Nr6a1⁺/ᶠˡˣ* compound line. Conditional deletion of *Nr6a1* was performed by intraperitoneal administration of Tam (60 mg/kg) into the pregnant dam at the specified embryonic stages. *Cdx2P:Nr6a1* transgenic embryos were generated by pronuclear injection by the Monash Gene Modification Platform. The full coding sequence of *Nr6a1* long isoform was synthesised with Not1 restriction enzyme site addition at the 5' and 3' ends by Biomatik, USA, sequence verified, digested, and cloned downstream of the Cdx2P promotor[53]. Two adult transgene-positive chimeras were recovered from multiple rounds of injection, however, a stable line could not be produced as one founder was infertile and one did not transmit the transgene. Thus, transient transgenic embryos were collected at E10.5 for tailbud RNA extraction and/or in situ hybridisation, and at E18.5 for skeleton preparation. Transgene copy number for each embryo was confirmed by droplet digital PCR.

### Genotyping

For routine genotyping, tail tissue was collected from postnatal pups and yolk sac collected from embryonic samples. Genotyping was performed by Transnetyx or in-house. For detection of *Nr6a1* conditional deletion via *TCreERᵀ²* and Tam administration at E7.5 or E8.5, digits of the left hindlimb were collected based on known pattern of expression of the T promoter[48,93]. To isolate the DNA, digit tissue was treated with Proteinase K (20 μg/ml) in 500 μL of Tris-HCl buffer (Composition: 100 mM Tris-HCl (pH8.0); 5 mM EDTA; 200 mM NaCl; 0.2% SDS) overnight at 56°C. The DNA was precipitated by adding equal volume of isopropanol and pelleted at 13,000 rpm. Next, the pellet was washed twice with 70% ethanol and eluted in nuclease-free water. At this point, genotyping was performed by standard PCR method using primers described in[46].

### ESCs maintenance

A mouse Bruce-4 ESC line was used for all in vitro differentiation assays. ESCs were routinely maintained on an in-house generated mitotically inactive primary mouse embryo fibroblast feeder layer in ES medium (81.8% Knockout DMEM (Gibco, 10829-018); 15% gamma irradiated foetal bovine serum (Gibco, 10101-145); 1% Pen/Strep (Gibco, 15140-122); 1% GlutaMAX-I (Gibco, 35050-061); 1% MEM NEAA (Gibco, 11140-040), 0.2% 2-Mercaptoethanol (Gibco, 21985-023)) supplemented with LIF (1000x, made in-house).

### Generation of *Nr6a1⁻ᐟ⁻* ESC line

*Nr6a1* null allele generation was performed by the self-cloning CRISPR/Cas9 (scCRISPR) method[59,60]. Two protospacers in *Nr6a1* exon4 were determined using the tool, CRISPOR (http://crispor.tefor.net/)[94]. The sgRNA oligonucleotides order for scCRISPR contains the protospacer (19-21 bp), flanked by sgPal7 (Addgene, 71484) homology (U6 promoter and sgRNA) sequence, total 60 bp. Three consecutive PCR steps were performed using three pairs of standard primer sets (sgRNA_HDRstep1-3) for preparing homology directed repair (HDR). ESCs used for electroporation were cultured and maintained as previously described without antibiotics. Once the ESCs were ready, they were pelleted and suspended in buffer R (ThermoFisher, MPK1025) at a density of 1*10⁷ cells/ml. 100 μl of the ESCs were then mixed with the scCRISPR construct (Composition: 10 μl elute of HDR PCR product; 1 μl sgPal7 (Addgene, 71484); 1 μl spCas9-BlastR (Addgene, 71489). ESCs were electroporated at pulse voltage 1,400 and width (ms) 3 for three cycles, and then plated on a 10 mm feeder-covered plate. 96 single colonies were selected and expanded independently in a 96-well plate. Multiple independent knockout lines were produced by this method and clones selected based on normal morphology. Clone B1 used in these studies harboured independent genomic deletions within exon 4 of each *Nr6a1* allele, one deleting 125 bp (Δ125) and the second 219 bp (Δ219) (Supplementary Fig. 8). Compared to the wildtype Nr6a1-long isoform protein product of 495 amino acids, Δ125 generates a truncated protein of 126 amino acids with no DNA binding and ligand-binding domain, while the in-frame deletion Δ219 allele encoded a protein of 422 amino acids with no DNA-binding domain, both supportive of a functional null[45].

### Gene cloning

*Nr6a1* isoform sequences were confirmed using gene amplification from E10.5 embryonic tissue. RNA was extracted (Macherey-Nagel) and cDNA prepared (Roche, 04897030001) using both random hexamers and oligodT primers. Sequence-specific primers were designed based on the version mm10 genome (UCSC) as detailed in Key resources table. Amplified sequences were cloned into pGEM-T easy vector (Promega) and used as the template for synthesising RNA riboprobes.

### Whole-mount in situ hybridisation

Whole-mount in situ hybridisation was performed as previously described[95] with minor modifications. Mouse embryos were dissected and placed in iced-cold PBS (Gibco, 14190-144). For embryos E9.5 or older, the brain's 4th ventricle was pierced to prevent probe trapping. Embryos were fixed in 4% paraformaldehyde (PFA) at 4°C, rocking

overnight. Fixed embryos were washed twice in PBT (Composition: 137 mM NaCl; 2.7 mM KCl; 10 mM $Na_2HPO_4$; 2 mM $KH_2PO_4$ in DEPC-$H_2O$ (pH7.4); 0.1% Tween) for 5 min and dehydrated into methanol using a graded MeOH/PBT series (25%, 50%, 75%, 100% MeOH) for 5-20 mins each with gentle rocking at room temperature (RT). To start in situ hybridisation, embryos were rehydrated into PBT using a graded MeOH/PBT series (75%, 50%, 25% MeOH) for 5-20 mins each and washed in PBT twice for 5 min with gentle rocking at RT. The embryos were then treated with 10 μg/ml of proteinase K in PBT as follows: E8.5 for 3.5 min, E9.5 for 8 min, E10.5 for 15 min and E12.5 for 25 min at RT with gentle rocking. The reaction was quenched by washing embryos twice in PBT for 5 min and the following postfix in 4% PFA with 0.2% glutaraldehyde for 20 min at RT. Embryos were then washed twice in PBT for 5 min at RT and transferred to hybridisation solution (Composition: 50% formamide; 5x SSC (pH4.5), 1% SDS; 50 μg/ml heparin; 50 μg/ml yeast tRNA (Sigma, R6750)) at 70°C with constant rocking for at least 2 hr before 1 μg/ml DIG-labelled riboprobe was added and incubated overnight. On the second day, the embryos were washed 3 times at 70°C in pre-warmed solution I (Composition: 50% formamide; 5x SSC (pH4.5); 1% SDS) for 30 min, followed by 3 washes at 65°C in pre-warmed solution II (Composition: 50% formamide; 2x SSC (pH4.5); 0.1% Tween-20) with gentle rocking. The embryos were then washed 3 times in TBST (Composition: 137 mM NaCl; 2.7 mM KCl; 25 mM Tris-HCl (pH7.5); 0.1% Tween-20) for 5 min at RT and blocked in TBST containing 10% heat-inactivated sheep serum (HISS) for 2 hrs at RT with gentle rocking. The embryos were then transferred to TBST containing 10% HISS and anti-DIG-AP antibody (Roche, 11093274910) at a dilution of 1:2000 overnight at 4°C with constant rocking. On the third day, the embryos were washed at least 5 times in TBST for 1 hr each wash at RT with gentle rocking and then overnight in TBST at 4°C. On the fourth day, embryos were equilibrated into NTT (Composition: 100 mM NaCl, 100 mM Tris-HCl (pH9.5); 0.1% Tween-20) by washing 3 times for 10 min each at RT with gentle rocking. To develop the colour, the embryos were incubated in BM purple (Roche, 11442074001) at RT. To stop the colour reaction, embryos were washed 3 times in PBT for 5 min, and postfixed in 4% PFA for 20 min at RT. The embryos were then washed 3 times in PBT for 5 min and stored in PBT at 4°C until photographing.

## Microarray

E9.0-E9.5 wildtype and $Nr6a1^{-/-}$ embryos were isolated and yolk sac DNA genotyped in parallel. Total RNA was extracted from each whole embryo using TRIzol reagent and processed through an RNeasy column (QIAGEN) using the RNA clean-up protocol. Concentration and quality of RNA were determined by spectrophotometer and Agilent bioanalyzer analysis (Agilent Technologies, Inc., Palo Alto, CA). For array analysis, labelled mRNA targets were prepared from 150 ng total RNA using MessageAmp III RNA Amplification Kit (Applied Biosystems / Ambion, Austin, TX) according to manufacturer specifications. Array analysis was performed using Affymetrix GeneChip GeneChip Mouse Genome 430 2.0 Arrays processed with the Gene-Chip Fluidics Station 450 and scanned with a GeneChip Scanner 3000 7G using standard protocols.

## Tailbud dissection for gene expression analysis

For E9.5 tailbud collection, all tissue at and caudal to the posterior neuropore was collected. For E10.5 tailbud collection, all tissue caudal to the last somite was collected. Dissected tissue was immediately placed into lysis buffer (Macherey-Nagel) on dry ice and stored at −80°C. RNA was extracted (Nucleospin RNA kit, Macherey-Nagel, 740955). The remaining part of each embryo was processed for whole-mount in situ hybridisation detection of *Uncx4.1* and accurate somite count determination, with embryo comparisons restricted to those within ±1 somite.

## Quantitative PCR using BioMark Fluidigm

100 ng RNA isolated from E9.5 and E10.5 tailbud tissue was used as template for cDNA synthesis performed using RT-Vilo (ThermoFisher). Quantitative PCR was performed using the 96*96 BioMark Fluidigm format. Raw Ct values were analysed using a modified version of the qPCR-Biomark script (https://github.com/jpouch/qPCR-Biomark) and normalised as previously described[96]. Only Ct values in the optimal range for the Biomark system of 6-25 were used for further analysis. All genes were first normalised against the mean raw Ct-values of five housekeeping gene probes yielding ΔCt values, then normalised against the wildtype condition yielding ΔΔCt values.

## Quantitative PCR using Roche Lightcycler

RNA isolated from one E9.5 tailbud or 1 mg RNA isolated from the in vitro cells was used as template for cDNA synthesis (Roche, 04897030001) using random hexamers and eluted in 200 μl and 100 μl $H_2O$ respectively. Each 10 μl PCR reaction reaction (5 μl SYBR Green I Master Mix (Roche, 04887352001); 2.5 μl $H_2O$; 2 μl cDNA; 0.25 μl 10 mM forward primer; 0.25 μl 10 mM reverse primer) was run on a Lightcycler 480 (Roche) using the program: 95°C for 10 s (1 cycle), 95°C for 10 s, 60°C for 15 s, 72°C for 10 s (45 cycles). Biological samples were processed in technical triplicate for each gene and the expression of housekeeping gene *Pol2a* used for normalising gene expression using the ΔΔCt method.

## RNA-seq

For in vivo sample RNAseq, tissue was dissected and RNA extracted as described above. RNA quality and quantity assessed using the Agilent Technologies BioAnalyser. Multiplexed RNAseq libraries, generated from 50 ng total RNA per sample, were prepared using Illumina TruSeq Stranded mRNA Sample prep (Protocol 15031047 Rev Oct 2013). 75 bp single-end (SE) sequencing was performed on the NextSeq500 Illumina platform. On average, 69 million SE reads were obtained for each library. For in vitro cell sample RNAseq, libraries were prepared by an in-house method. The index is added during initial pA priming and pooled samples amplified using template switching oligos. P5 was added by PCR and P7 by Nextera transposase. The final library structure is as follows: P5-Rd1->8 bp index-10bpUMI-pA then cDNA < -Rd2 primer i7 index P7. Paired end sequencing was performed on the NextSeq550 Illumina platform for 19 bp forward reads of index and 72 bp reverse reads of cDNA. On average, 20 M raw reads were obtained for each library.

## Skeleton preparation and imaging

Skeletal preparation was performed on E16.5 and E18.5 embryos as previously described[97]. Embryos were skinned, then incubated for 2 days at RT with constant rocking in each of the following solutions: 95% ethanol, 100% acetone, staining solution (Composition: 15 mg alcian blue and 5 mg alizarin red S in 70% ethanol with 0.5% glacial acetic acid). To clear the skeletons, stained embryos were washed in 1% KOH at RT with constant rocking for 2-5 days, then transferred into glycerol using a graded glycerol/1% KOH series (25%, 50%, 75%, 100% glycerol) for 24 hr in each solution with gentle rocking at RT. Finally, the embryos were transferred in 100% glycerol with 0.02% sodium azide for long-term storage. Images were acquired with a Vision Dynamic BK Lab System at the Monash University Paleontology Lab. Images were taken with a Canon 5d MkII with a 100 mm Macro lens (focus stop 1:3/1:1). Multiple images were taken to extend the focal depth, and stacked in ZereneStacker using the PMax algorithm.

## NMP differentiation of ESCs

NMP differentiation was based on the published protocols[20,56] with minor modifications. In preparation for differentiation, ESCs were

feeder-depleted and plated in gelatin-coated (0.1% Sigma, G1890-100G) 6-well plates (Falcon, 353046) at a density of $8\times10^3$ cells/cm$^2$ in ES medium. On D0 of differentiation media was changed to N2B27 media (Composition: 49.5% Advanced Dulbecco's Modified Medium F-12 (Gibco, 12634028); 49% Neurobasal medium (Gibco, 21103049); 0.5% N2-supplement (Gibco, 17502001); 1% B27-supplement (Gibco, 17504044)) supplied with 1x Glutamax (Gibco, 17504044), 40 μg/ml BSA Fraction V (Gibco, 15260037) and 100 mM 2-Mercaptoethanol (Gibco, 21985-023)) supplemented with 10 ng/ml hFGF-2 (Miltenyi Biotec, 130-104-925). The media was further supplemented with 5 μM CHIR99021 (StemMACS, 130-103-926) on D2 and with or without 50 ng/ml hGdf11 (130-105-776, Miltenyi Biotec) on D3. Throughout differentiation, the medium was refreshed every 24 h. NMP identity at D3 was routinely confirmed by co-expression analysis of *Sox2* and *T/Bra* using immunofluorescence.

### Flow cytometry analysis

Samples for flow cytometry were isolated and analysed from individual embryos. All tissue posterior to the first visible somite condensate was dissected from E9.0 WT or *T-Cre*$^{ERT2}$;*Nr6a1*$^{flox/flox}$ (Tamoxifen-treated @ E7.5) embryos in cold PBS and dissociated for 20' in 120 μL Accutase at 37 °C. Single cell suspension was achieved by pipetting up and down with a P200 low bind pipette tip before adding 800 μL of PBS + 2% Foetal Bovine Serum (FBS) and pipetting up and down with a P1000. This single-cell suspension was then filtered through a 70 μm mesh cap and cells pelleted by centrifugation at 400 g. Cells were stained using the viability dye Zombie Aqua (BioLegend), fixed and permeabilised using the True Nuclear Transcription Factor Buffer Set (BioLegend), both following manufacturer's instructions. Cells were incubated with primary antibodies (rat anti-Sox2, 1:200, eBiosciences; rabbit anti-T/Bra, 1:200, Abcam) for 30 minutes in PERM buffer in the dark at room temperature (RT), then washed once in PERM buffer. Secondary antibodies (goat anti-rat-AlexaFlour555, Invitrogen; goat anti-rabbit-AlexaFlour 647, Invitrogen), were incubated in PERM buffer for 30 minutes in the dark at RT, washed once in PERM buffer followed by a second wash in PBS + 2%FBS. Cells were pelleted by centrifugation at 400 g, resuspended in PBS + 2%FBS and 1 mg/ml Propidium iodide and analysed by flow cytometry on a LSRFortessa (BD). A minimum of 8,000 and a maximum of 16,000 events were recorded per sample. To set gates, all non-tailbud tissue was pooled and stained for 1) both secondary antibodies, 2) anti-Sox2 primary and secondary alone and 3) anti-T/Bra primary and secondary alone. Experimental cells were first gated to remove debris, then gated based on size to select for single cells, and finally, cells that had taken up Propidium iodide were excluded leaving single viable cells for analysis of Sox2 and T/Bra expression as depicted in Supplementary Fig. 13.

### Quantification and statistical analysis

**Microarray analysis.** Microarray CEL files were analysed using RMA[98] and limma[99] in the R statistical package. Following alignment and annotation of genes, the Microarray data was filtered to export differentially expressed genes with an adjusted p-value or false discovery rate of 0.05 or lower and a fold change less than or equal to −2 and greater than or equal to 2 (log$_2$FoldChange less than or equal to −1 and greater than or equal to 1). Genes were then sorted to extract the top 50 most downregulated genes, which were analysed via DAVID[100,101] and ToppGene[102] utilising standard settings provided in each program. The results were then exported into tables in Excel and the bar graph of biological processes affected by the top 50 downregulated genes was generated using -log10(p-value) parameters and plotted using PRISM.

### Vertebral formulae—*CMVCre*$^{ERT2}$ analysis

For statistical analysis and data visualisation of vertebral numbers, Graph Pad Prism 9.3.1 (471) was used. A two-tailed T-test was employed. Error bars represent the mean with standard deviation.

### Vertebral formulae—*TCre*$^{ERT2}$ analysis

For statistical analysis and data visualisation of vertebral numbers, R-package "Data Analysis using Bootstrap-Coupled ESTimation" (dabestr) was used, and the background of the methods were previously described[103]. To determine mean differences to the respective shared control, 5000 bootstrap samples were taken and the confidence interval was bias-corrected and accelerated. In the visualisation, 95% confidence interval is indicated by the ends of the vertical error bar and the sampling error distribution is diagrammed as a grey filled curve. The codes are available at https://github.com/ACCLAB/dabestr.

### Quantitative PCR using Roche Lightcycler

All genes were first normalised against the mean raw Ct-values of housekeeping gene, *Pol2a*, yielding ΔCt values. Following normalisation of gene expression using the ΔΔCt method, statistical analysis was performed using the Wilcoxon test.

### RNAseq analysis

All sequencing reads were aligned to the reference genome using STAR aligner[104]. Only the genes with counts, which are greater than 10, and with CPM, which are greater than 2 in two biological replicates were used for further analysis. Differential gene expression between control and mutants were performed using edgeR[105,106]. Here we used false discovery rate (FDR), the adjusted p-value, to display significance of the differential gene expression between the controls and the mutants. Genes with a FDR < 0.05 were considered to be significantly differentially expressed in the mutants.

### Resource availability

Further information and requests for resources and reagents should be directed to and will be fulfilled by Edwina McGlinn (edwina.mcglinn@monash.edu).

### Reporting summary

Further information on research design is available in the Nature Portfolio Reporting Summary linked to this article.

## Data availability

The data that support this study are available from the corresponding author upon request. The raw microarray data generated in this study have been deposited in the Gene Expression Omnibus (GEO) and are publicly available using the accession number GSE166458 (CEL datasets). Original data relating to the microarray experiment can also be accessed from the Stowers Original Data Repository LIBPB-1647. The raw in vitro and in vivo RNAseq data generated in this study have been deposited in the GEO and are publicly available using the accession number GSE180427. This includes raw data for graphs presented in this study. Source data are provided with this paper.

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

## Acknowledgements

The authors acknowledge the excellent technical assistance provided by Monash Gene Modification Platform, Monash Animal Research Platform and the MHTP Medical Genomics Facility, Karin Zueckert-Gaudenz and Allison Peak for their microarray technical expertise and to Dr Chris Seidel for bioinformatic analysis support. We also greatly appreciate Melissa Childers and the Laboratory Animal Services Facility at the Stowers Institute for their care and maintenance of our mice. The *Gdf11* mutant mouse line was kindly provided by Professor Se-Jin Lee and the *Nr6a1*^flx line was kindly provided by Professor Austin Cooney. Y.-C.C. and G.M.H. were supported by an Australian Government Research Training Program Scholarship. A.A. was the recipient of an American Association for Anatomy Post-Doctoral Fellowship. J.M.P. is supported by an Australian Research Council Future Fellowship. This work was supported by Australian Research Council Discovery Project DP180102157 to E.M. and J.M.P. Research in the Trainor laboratory is supported by the Stowers Institute for Medical Research. The Australian Regenerative Medicine Institute is supported by grants from the State Government of Victoria and the Australian Government.

## Author contributions

E.M. and Y.-C.C. conceived the study and wrote the manuscript. Y.-C.C., S.F.L.W., G.M.H., N.A.S., E.L.M., A.A., V.G., J.M. and P.T. performed experiments and analysed data. J.S. and J.M.P. provided advice on experimental design and analysed the RNA-seq data. All authors have commented on the manuscript and approve submission of the manuscript.

## Competing interests

The authors declare no competing interests
