## [Peer Review File · Nature Communications]

REVIEWER COMMENTS

Reviewer #1 (Remarks to the Author):

Chang and colleagues present a study demonstrating a novel role for the orphan nuclear receptor Nr6a1 as a general regulator of thoraco-lumbar segment number and identity, with roles also proposed in segmentation itself and in promoting neural fate choice in neuromesodermal progenitors. This is an important piece of work that significantly moves forward our understanding of the regulation of groups of axial segments, and together with the related paper, shows that Nr6a1 plays an opposing role to a second master regulator, Gdf11, which controls sacro-caudal axial identity, and part of this involves its prevention of precocious activation of posterior Hox genes. Overall the data is high quality and convincing, and my comments concern only some relatively minor aspects of the arguments.

Data in figure 2 is presented to support the idea that lumbar vertebrae are transformed almost wholesale into sacral ones. It would help to convince the reader that these are not just aberrant vertebrae if the data were presented alongside images of normal sacral vertebrae (NB the figure legend states that the arrows are green but the panel shows red arrows). If possible, wholemound in situ hybridisation for Hox gene expression characteristic of sacral not lumbar identity would support this statement. Data in figure 3 supports the case that Nr6a1 stalls expression of the most posterior (PG13) Hox genes, but expression is measured in the tail bud, not the segments themselves so don't directly show the vertebral identity.

The data in fig 3a is additionally a bit tantalising. The repressive effect of Nr6a1 on Hox PG13 genes seems convincing, especially when corroborated by the in situs in B. But because there seems to be quite a lot of variation in expression levels of the other Hox genes between embryos (particularly wildtype ones), many effects that might otherwise have been significant, aren't. I wonder whether any factors such as exact number of vertebrae could explain this variation; or alternatively, could increasing the sample size help to determine whether there is a significant effect on any of the PG10-12 Hox genes? This might help decide whether Nr6a1 specifically affects PG13 genes or more generally affects the posterior Hox genes.

The data supporting the prolonged thoraco-lumbar regional identity after prolonged Nr6a1 expression is much more robust, although it too could perhaps benefit from wholemound in situ hybridisation for potentially affected Hox genes. Overall the data supports the hypothesis that the trunk depends on Nr6a1 and suppression of Gdf11, while the tail needs Gdf11. I think however that the authors should perhaps revisit the wording of the claim that 'two modules affect axial elongation' (e.g. last paragraph of the introduction), as the data is silent about the cervical

vertebrae, which contribute to axis elongation, and also there is no suggestion that these are the only two modules affecting axis elongation- presumably Nr6a1/Gdf11 are high-level but below these are modules that subdivide the axis into thoracic, lumbar, sacral and caudal identities, and fix the number and identity of individual segments.

My biggest doubt over the data concerns the claim that Nr6a1 affects the balance of neural versus mesodermal differentiation. There seems to be no indication from the phenotypes of either over- or underexpressing Nr6a1 that the neural/mesodermal ratio is affected- is this just because this hasn't been directly examined? Would transverse sections of the embryos shown in Fig2f show an enlarged mesodermal domain compared to the neural tube? Moreover, the markers used to support the statement that Nr6a1 promotes neural differentiation of NMPs are a bit problematic. While one of the affected genes, Sox1, is truly a neural marker, many of the others are additionally expressed in lateral, intermediate or extraembryonic mesoderm at the stages examined (I checked Pijuan-Sala et al Nature 2018 for expression of the in vitro markers and MGI Jax expression database for the E10.5 marker set). The mesodermal markers upregulated/downregulated are primarily paraxial, raising in my mind the possibility that Nr6a1 may be affecting not a neural/mesodermal but a paraxial versus lateral mesoderm decision. It's intriguing but not really commented on that, consistent with an effect on lateral plate mesoderm, there seems to be a reduced size of hindlimbs after Nr6a1 downregulation (Fig2b,f) but not in the Nr6a1 upregulated embryos, eg those shown in Fig4a.

Reviewer #2 (Remarks to the Author):

Axial elongation is a conserved mechanism driving the generation of the body plan in vertebrates. Neuromesodermal progenitors (NMPs) are fueling the process of axial elongation by giving descendants to the spinal cord and associated somites. Whereas several studies using rodent, chick embryos, and most recently pluripotent stem cells, have shed light on the role of specific gene regulatory networks and signals that interact to regulate posterior body patterning and axial growth from NMPs, there are still several open questions. In this study Chang et al, using an elegant combination of mouse genetics and mouse embryonic stem cell differentiation experiments they identify Nuclear receptor subfamily 6 group A member 1 (Nr6a1) as a master regulator of trunk elongation.

Nr6a1 is widely expressed in the early mouse embryo but it has been also shown that it exhibits a temporally restricted expression in NMPs at e8.5. The authors build on this interesting observation and perform mechanistic studies to understand the role of Nr6a1 in axial elongation. Analysis of conditional gain and loss of function experiments in mouse embryos provides very convincing evidence that Nr6a1 acts as a rheostat to control the number of vertebra in the trunk region. They further study the levels of expression of Nr6a1 in single and compound mutants of miR196 and

GDF11 embryos that have an increased number of trunk vertebrae and observe a clear correlation between the number of vertebrae and the levels of Nr6a1 expression which further supports the role of Nr6a1 in the trunk elongation process. They further find that Nr6a1 is sufficient to control the temporal activation of trunk Hox genes and prevents the expression of more posterior Hox genes. Finally, using embryonic stem cell differentiation experiments they analyse the role of Nr6a1 in the cell lineage choice of axial progenitors and identify that Nr6a1 favours the neural differentiation.

Overall, the manuscript is clearly written and very well presented. The data provide compelling evidence that there are two distinct processes that control the trunk and tail modules of axial elongation and that Nr6a1 is a master regulator of the trunk growth. The reviewer is happy to see that the findings are in agreement with the temporal changes in the transcriptional signature of NMPs that is essential to support this very dynamic process during embryonic development.

I have only a few comments to bolster the findings of the paper regarding the role of Nr6a1 in axial elongation. These are summarized below:

- Nr6a1 is expressed in NMPs of wild type embryos at e8.5 and is downregulated at e9.5. However, it is not clear if the expression of Nr6a1 is maintained in the NMPs of the GDF^{-/-} for longer? The authors should include immunofluorescence data of Nr6a1 with T/Bra/Sox2 at e8.5, e9.5, e 10.5

- Is the expression of Raldh2 and Cyp26a1 affected in the posterior part of the Nr6a1^{-/-} embryos?

- The authors should include immunofluorescence data of T/Bra/Sox2 to prove the efficient generation of NMP cells at day 3 in the absence of Nr6a1.

- Have the authors compared the transcriptional signature of the Nr6a1^{-/-} d3 NMPs to the in vivo e8.5 and e9.5 NMPs? Are the Nr6a1^{-/-} NMPs more similar to e9.5 NMPs?

- Is the expression of Nr6a1 maintained longer in NMPs treated with TGFb inhibitor between d3 and d4?

Minor points

- In Figure 3B there are no red arrows that indicate the region containing the NMP cells as described in the legend.

- In the title of Figure S9, there is a typo on ESCs

Reviewer #3 (Remarks to the Author):

The paper by Chang et al. addresses the role of the Nuclear receptor subfamily 6 group A member 1 (NR6A1) in somitogenesis and trunk elongation. Nr6a1 is broadly expressed in the developing mouse embryo, including the neuromesodermal progenitors (NMPs). Single nucleotide polymorphisms (SNPs) within Nr6a1 in mammals have been associated with increased trunk vertebral number. Based on these observations, the Authors hypothesize that Nr6a1 has a role in vertebrae formation in mice.

To address this question, they analyzed Nr6a1 expression via in situ hybridization in mouse embryos. They performed skeletal preparation analyses to assess the skeletal defects of the Nr6a1 loss of function and overexpressing alleles. They analyzed the paraxial mesoderm defects looking at the changes in somites identity by studying the Hox-code expression via RNAseq in the mutant embryos. Finally, the authors established an in vitro cell culture system to corroborate the findings observed in the developing embryos.

The authors reported that Nr6a1 regulates posterior Hox genes expression in mouse tailbuds, and thus has a critical role in controlling the number of lumbar and sacral vertebrae. Additionally, they proposed that Nr6a1 plays an essential role in cell lineage choice of axial progenitors regulating the balance between neural and mesodermal fate. Finally, they suggested that Nr6a1 transcriptional regulation is mediated by the priming activity of SMAD2/3 at the epiblast state.

While the hypothesis that changes of vertebrae identity in Nr6a1 mutants is linked to the shift in somite identity and altered Hox code is quite intriguing, the experiments included in the current version manuscript do not fully support the claims and hypothesis.

The manuscript focused mainly on the regulation of Hox genes expression by NR6A1. HOX are critical key players of axial elongation and vertebrae patterning. However, the mechanism by which Nr6a1 controls posterior Hox genes is not fully elucidated.

Regarding the methodology employed, the authors generated two new mouse alleles, overexpressing Cdx2::Nr6a1 and new Nr6a1 mutant ES cell lines. However, no major cutting-edge technologies are used or implemented in the current manuscript.

In the opinion of this reviewer, there are no molecular methods included in this manuscript that shed light into an indirect or direct regulation of Hox genes by Nr6a1, yet it is a major claim in the paper. Dissecting the mechanisms by which Nr6a1 controls Hox expression is mandatory to

understand the vertebrae phenotypes observed in the Nr6a1 mutants, and it is a key missing element of the paper.

Major critiques:

The authors reported that the expression and regulation of Nr6a1 correlate with trunk formation in the mouse. Although they said temporal enrichment in NMPs and somites between E9.5 and E10.5, Nr6a1 appears to be broadly expressed, including in the epiblast stages. What provides specificity to its function in NMPs and somites? The authors should use immunofluorescence (IF) to assess NR6A1 protein activity in the tailbud and test whether it has specific nuclear staining in the caudal lateral epiblast, NMP territories, or somites.

The sentence: “these data revealed a dynamic pattern of Nr6a1 expression that correlates with axial progenitors of the trunk region and one that is terminated by the synergistic action of miR-196 and Gdf11 signaling” should be clarified.

Overall it is not clear the rationale for the experiments in Fig 1C-D. What is the author’s intention? Do they want to study the genetic interaction between Nr6a1 and Gdf11? If this is the case, why? And why is it addressed at this point before knowing if Nr6a1 has a role in axial elongation?

The authors should revise the manuscript and condense the panels of Fig1 and 2 in one Figure. The main message of Fig1 should be the expression and phenotype of the mutant Nr6a1 embryos.

While it is clear that there are different numbers of lumbar and sacral vertebrae, it is not so clear if this phenotype is a result of a change in somite identity or an aberrant number of somites (Fig2F). The somites should be counted using reference points, both the forelimb and hindlimb (like in McPerson et al. 1999). A table providing statistical significance of the observed difference between controls and mutants should be included.

The authors claimed that Nr6a1 inhibits posterior Hox gene expression in the tailbud.

There is a differential expression of Hox genes, but that does not necessarily mean that Nr6a1 inhibits Hox expression in the tailbud. The effect on the Hox could be a result of a general developmental delay.

To prove that Nr6a1 inhibits posterior Hox gene expression, the underlying mechanism of inhibition should be clearly addressed.

In other words, how does Nr6a1 regulates Hox genes in the tailbuds? Are the 4 Hox paralogs the directed targets of NR6A1? ChIP addressing the direct targets of NR6A1 using either embryonic tissues or cultured cells is critical and mandatory to address this point.

In addition to the Hox genes, which other genes are differentially regulated in the Nr6a1 mutant tail buds? Maybe the effect on Hox genes is indirect and Nr6a1 controls important upstream regulators like WNT.

Finally, what provides specificity to a broadly expressed gene like NR6A1 in the tailbud. How can it target specifically the posterior Hox genes?

The authors should consider generating a NR6A1-GFP or TAG allele, valid for the ChIP experiments (if a ChIP grade antibody is not available), and immunoprecipitate NR6A1 to identify potential binding partners that could provide information about the DNA-binding specificity.

The authors hypothesized that the panel of Hox genes expression in Fig 3A is associated with the NMPs. Still, Hox genes are expressed in many other cell types, including neurons. Colocalization with NMP markers or the use of the scRNA-seq is required to unequivocally attribute the precocious expression of posterior Hox genes to NMPs. A similar consideration can also be raised for Fig 3B.

Performing scRNAseq analysis will significantly help with the attribution of Hox genes to the NMP cluster and dissect the phenotypes.

To test if Nr6a1 clearance is essential for the trunk-to-tail transition, the authors generated a new mouse allele where Nr6a1 is under the control of the Cdx2 promoter. So that Nr6a1 will be expressed at E10.5 in the tailbud when Nr6a1 is no longer present. Cdx2::Nr6a1 showed an increased number of thoraco-lumbar vertebrae.

If NR6A1 controls posterior Hox gene expression, why the thoracic vertebrae that are controlled by middle Hox are affected? This observation raises again the question of whether the Hox genes are the primary targets. The phenotype of the thoracic vertebrae is strong while the deregulation of thoraco-lumbar Hox genes are modest. So what does NR6A1 really do?

The authors claim that "Nr6a1 alone is sufficient to control the temporal progression of Hox activation of all 4 Hox clusters." – this sentence should be toned down because a direct link between Nr6a1 and Hox gene regulation has to be demonstrated. Furthermore, NR6A1 cannot work alone, but it has to work together with cell-specific factors. Otherwise, a similar effect on Hox genes should be seen in all the territories where Nr6a1 is expressed.

In the differentiation system (Fig6), the authors do not consider the cell populations' heterogeneity during differentiation. At d2, they listed the generation of EpiSCs; what is the evidence that those

are EpiSCs? Or that at d3 there are NMPs? The only markers used to attribute cell identity are the Hox genes. As discussed in Figure 3, Hox genes are expressed in many other cell types. Colocalization with NMP markers or the use of the scRNA-seq is required to unequivocally attribute the precocious expression of posterior Hox genes to NMPs.

What do you expect to find at d4?

The authors suggested that Nr6a1 promotes neural fate within bipotent NMPs. Could a neural phenotype be identified in the Nr6a1 mutant embryos? Is the neural tube duplicated like in the Tbx6 mutants?

Where are these progenitors localized in the embryos?

NMPs fated to neuronal and mesodermal fate should be identified in the embryos using specific lineage markers. Neuronal and mesodermal progenitors should be counted and compared to the controls.

In Fig 7, why the comparison between in vivo and in vitro systems are done with two different mutated alleles?

The authors suggested that Nr6a1 is regulated by its antisense transcript, showed by the opposite pattern of expression during differentiation. They proposed a potential priming effect of SMAD2/3 in the epiblast that regulates Nr6a1^{los} and, in turn, Nr6a1. Given these observations are purely speculative, they should mention them in the Discussion session only.

To prove that Nr6a1^{los} regulates Nr6a1 via SMAD2/3. The authors should delete the SMADs binding sites of Nr6a1^{los}, see the effect on Nr6a1 expression, and test if it recapitulates the phenotype?

No precise details about the statistical methods used have been reported. This information is also missing in the Material and Methods.

Western blot proving the absence of NR6A1 in ES cell lines should be provided. Given the mutant allele has been generated by deleting the DNA binding motif, a truncated protein can be produced, raising the possibility of a dominant-negative effect.

The Discussion is nicely written, but it sounds as Evo-devo review and it is unlinked with the findings described in the Results.

Reviewer #1 (Remarks to the Author):

Chang and colleagues present a study demonstrating a novel role for the orphan nuclear receptor Nr6a1 as a general regulator of thoraco-lumbar segment number and identity, with roles also proposed in segmentation itself and in promoting neural fate choice in neuromesodermal progenitors. This is an important piece of work that significantly moves forward our understanding of the regulation of groups of axial segments, and together with the related paper, shows that Nr6a1 plays an opposing role to a second master regulator, Gdf11, which controls sacro-caudal axial identity, and part of this involves its prevention of precocious activation of posterior Hox genes. Overall the data is high quality and convincing, and my comments concern only some relatively minor aspects of the arguments. We thank the reviewer for the overall positive comments

Data in figure 2 is presented to support the idea that lumbar vertebrae are transformed almost wholesale into sacral ones. It would help to convince the reader that these are not just aberrant vertebrae if the data were presented alongside images of normal sacral vertebrae (NB the figure legend states that the arrows are green but the panel shows red arrows).

We agree this was not as clear as it should have been. We have incorporated additional images focusing on the lumbo-sacral region of one WT and two *TCre^{ERT2};Nr6a1^{flx/flx}* embryos (Figure 2B) and adjusted the text as follows:

Pg 11. Removal of the pelvic bones in a WT embryo allowed visualisation of the normal winged-shaped transverse process on sacral elements S1 and S2 which at this stage are becoming fused laterally, and the rostrally-pointing transverse process of lumbar elements L4, L5, L6 (Figure 2B). In *TCre^{ERT2};Nr6a1^{flx/flx}* embryos, positioning of the normal (albeit malformed) sacral region was appropriately spaced relative to the last formed thoracic element, defined by the presence of adjacent rudimentary pelvic/hindlimb bones (blue S, Figure 2B). Rostral to this, the transverse processes of all intervening post-thoracic vertebral elements assume a flattened morphology which, in numerous embryos, show lateral fusion between adjacent elements (red S, Figure 2B; asterisk marks lateral fusions). These unique characteristics of sacral morphology indicated an almost complete transformation of lumbar to sacral identity in *TCre^{ERT2};Nr6a1^{flx/flx}* embryos.

If possible, wholemount in situ hybridisation for Hox gene expression characteristic of sacral not lumbar identity would support this statement. Data in figure 3 supports the case that Nr6a1 stalls expression of the most posterior (PG13) Hox genes, but expression is measured in the tail bud, not the segments themselves so don't directly show the vertebral identity.

We thank the reviewer for this suggestion and have performed whole mount in situ hybridisation for *Hoxd11*, since Hox11 paralogs are known regulators of sacral morphology (Wellik and Cappechi, *Science*, 2003). The results nicely corroborate the morphological assessment and we have adjusted the text as follows:

Pg 11. To corroborate this morphological assessment, we performed whole mount in situ hybridisation for *Hoxd11*, a known regulator of sacral identity (Wellik and Cappechi, 2003). At E10.5, we see a striking shift in the rostral boundary of *Hoxd11* in *TCre^{ERT2};Nr6a1^{flx/flx}* embryos compared to somite-matched WT embryos (Figure 3D). This shift was specific to the paraxial mesoderm, with ectopic *Hoxd11* expression observed in at least 6-7 somites rostral to the normal boundary of expression, correlating well with the extent of skeletal transformations observed (Figure 2B).

The data in fig 3a is additionally a bit tantalising. The repressive effect of Nr6a1 on Hox PG13 genes seems convincing, especially when corroborated by the in situs in B. But because there seems to be quite a lot of variation in expression levels of the other Hox genes between embryos (particularly wildtype ones), many effects that might otherwise have been significant, aren't. I wonder whether any factors such as exact number of vertebrae could explain this variation; or alternatively, could increasing the sample size help to determine whether there is a significant effect on any of the PG10-12 Hox

genes? This might help decide whether Nr6a1 specifically affects PG13 genes or more generally affects the posterior Hox genes.

We apologise for this oversight. To control for expression variation due to stage differences, we only ever perform comparisons between somite-matched embryos. Specifically, tailbuds were collected for qPCR analysis and the remainder of the embryo stained for the somite marker *Uncx4.1* allowing precise somite counting. We have included this data as a supplementary figure (Figure S5) and updated the main text and methods section as follows.

Pg. 12. Indeed, the expression of several posterior Hox genes were significantly up-regulated in TCreER^{T2};Nr6a1^{flx/flx} tailbuds compared to somite-matched (+/-1) controls,

Pg. 53. The remaining part of each embryo was processed for whole-mount in situ hybridisation detection of *Uncx4.1* and accurate somite count determination, with embryo comparisons restricted to those within +/- 1 somite.

We agree with the reviewer that the PG13 genes appear to be more strongly affected in this experiment. In this instance, we believe this is most likely because precocious/heightened activation would be more straightforward to detect for a gene that is just beginning to be turned on (as is the case for PG13s at E9.5) versus a levels difference of a gene that is already robustly turned on (PG10-12s). Our collective data supports the effect of Nr6a1 broadly spanning posterior *Hox* clusters, given the cumulative changes in posterior *Hox* genes following in vivo Nr6a1 gain-of-function (Figure 5C) and *in vitro* Nr6a1 loss-of-function (Figure 7A and 6B).

The data supporting the prolonged thoraco-lumbar regional identity after prolonged Nr6a1 expression is much more robust, although it too could perhaps benefit from wholemount in situ hybridisation for potentially affected Hox genes.

We have performed whole mount in situ hybridisation for *Hoxd12* at E10.5, a time point where robust *Hoxd12* expression is seen in the tail of wildtype embryos, with the results nicely corroborating the RNAseq analysis.

We have updated the text as follows:

Pg. 15. We went on to assess the expression of *Hoxd12* by whole mount in situ hybridisation in E10.5 *Cdx2P:Nr6a1* embryos, revealing a general reduction in expression throughout the entire tail compared to WT somite-matched embryos, with clear reductions observed in tailbud mesoderm and somitic tissues (Figure 5D).

Overall the data supports the hypothesis that the trunk depends on Nr6a1 and suppression of Gdf11, while the tail needs Gdf11. I think however that the authors should perhaps revisit the wording of the claim that 'two modules affect axial elongation' (e.g. last paragraph of the introduction), as the data is silent about the cervical vertebrae, which contribute to axis elongation, and also there is no suggestion that these are the only two modules affecting axis elongation- presumably Nr6a1/Gdf11 are high-level but below these are modules that subdivide the axis into thoracic, lumbar, sacral and caudal identities, and fix the number and identity of individual segments.

We agree with the reviewer on our over-simplification, and have modified the introduction as follows
Pg 7. at least two modules

My biggest doubt over the data concerns the claim that Nr6a1 affects the balance of neural versus mesodermal differentiation. There seems to be no indication from the phenotypes of either over- or underexpressing Nr6a1 that the neural/mesodermal ratio is affected- is this just because this hasn't been directly examined? Would transverse sections of the embryos shown in Fig2f show an enlarged mesodermal domain compared to the neural tube? Moreover, the markers used to support the statement that Nr6a1 promotes neural differentiation of NMPs are a bit problematic. While one of the affected genes, *Sox1*, is truly a neural marker, many of the others are additionally expressed in lateral, intermediate or extraembryonic mesoderm at the stages examined (I checked Pijuan-Sala et al Nature 2018 for expression of the in vitro markers and MGI Jax expression database for the E10.5 marker set). The mesodermal markers upregulated/downregulated are primarily paraxial, raising in my mind the possibility that Nr6a1 may be affecting not a neural/mesodermal but a paraxial versus lateral mesoderm

decision. It's intriguing but not really commented on that, consistent with an effect on lateral plate mesoderm, there seems to be a reduced size of hindlimbs after Nr6a1 downregulation (Fig2b,f) but not in the Nr6a1 upregulated embryos, eg those shown in Fig4a.

A similar point was also raised by reviewer 3 and we have performed additional analysis to strengthen this section. Please note, we have chosen to focus on the very early neural vs mesodermal progenitor cell fate choice, rather than the later stages of neural tube and somite elaboration. As the reviewers points out, we do not see dramatic changes in differentiated neural vs mesodermal tissues (eg. ectopic neural tube), and as these differentiating tissue are quite dysmorphic in *Nr6a1*^{-/-} embryos (especially the somites) it would be very challenging to infer direct vs indirect causes of any changes in differentiated tissue size or shape observed. Nonetheless, signals can have more subtle effects on early neural vs mesodermal fate choice and this is what we have focused on.

Specific response to Reviewer 1:

We agree that genes are often deployed across many tissue types, and that some of the neural- or mesoderm-enriched genes assessed may not be unique to these tissues, particularly as they differentiate. However, we used the gene lists from Gouti et al., Dev Cell 2017 as they represent the genes which differentiate neural progenitor cell (NP) and Mesoderm progenitors (MPs) away from NMPs within the in vivo tailbud tissue (our tissue and our question of interest). We have added the following text to clarify that point, and highlight the more robust lineage markers that support a potential bias in fate choice rather than using these lists to conclusively define tissue types.

Pg. 20. While some of these lineage markers assessed are not uniquely restricted to these lineages, particularly at differentiation stages, the inclusion of key lineage markers of tailbud progenitor stages, such as Sox1 (NP) and Tbx6 (MP) supported further investigation of a potential bias in lineage choice in vivo.

Importantly, to corroborate the molecular data presented in Figure 8A-B, we have performed Flow cytometry analysis of progenitor cell populations in the E9.0 wildtype and *TCre^{ERT2};Nr6a1^{flx/flx}* tailbuds. This new data supports Nr6a1 having a minor role in this early fate choice, and we have modified the text as follows:

Pg. 20. To this end, we performed Flow cytometry analysis of E9.0 wildtype and *TCre^{ERT2};Nr6a1^{flx/flx}* tailbuds, to quantify any shift in the numbers of T/Bra⁺; Sox2⁺ (NMPs), T/Bra^{neg}; Sox2⁺ (NP) and T/Bra⁺; Sox2^{neg} (MPs). The time point is important, since it is when cells that will go on to form the thoracolumbar region - the axial region that is strongly dependent on Nr6a1 function - are transiting through the PSM. While we observed no difference in the number of NMPs between WT and *TCre^{ERT2};Nr6a1^{flx/flx}* tailbuds, we saw a significant decrease in NPs and increase in MPs following loss of Nr6a1 (Figure 8C), consistent with the molecular biases observed (Figure 8A,B).

We have adjusted the summary sentence as follows:

Pg.20. Together, these results identify Nr6a1 as a novel intrinsic factor essential for that has a minor role in regulating the balance between neural and mesodermal fates within NMPs during axial elongation.

The reviewer is correct in observing a reduced hindlimb size in *TCre^{ERT2};Nr6a1^{flx/flx}* embryos which, while beyond the scope of this manuscript, is an active area of current research in our lab. We have mentioned the observation of rudimentary hindlimb bones in the main text (please see pg 11), and discussed that the cell-autonomous Nr6a1 actions are likely to extend across multiple tissue/germ layers (please see pg 28).

Of note, Tam activation of *TCre^{ERT2}* in E7.5 embryos has previously been shown to function in lateral plate mesoderm at forelimb level, suggesting that that the hindlimb-specific defects of *TCre^{ERT2};Nr6a1^{flx/flx}* embryos may not relate to global LPM lineage choice defects but rather a axially-restricted trunk defect, but this under current investigation.

Reviewer #2 (Remarks to the Author):

Axial elongation is a conserved mechanism driving the generation of the body plan in vertebrates. Neuromesodermal progenitors (NMPs) are fueling the process of axial elongation by giving descendants to the spinal cord and associated somites. Whereas several studies using rodent, chick embryos, and most recently pluripotent stem cells, have shed light on the role of specific gene regulatory networks and signals that interact to regulate posterior body patterning and axial growth from NMPs, there are still several open questions. In this study Chang et al, using an elegant combination of mouse genetics and mouse embryonic stem cell differentiation experiments they identify Nuclear receptor subfamily 6 group A member 1 (Nr6a1) as a master regulator of trunk elongation. Nr6a1 is widely expressed in the early mouse embryo but it has been also shown that it exhibits a temporally restricted expression in NMPs at e8.5. The authors build on this interesting observation and perform mechanistic studies to understand the role of Nr6a1 in axial elongation. Analysis of conditional gain and loss of function experiments in mouse embryos provides very convincing evidence that Nr6a1 acts as a rheostat to control the number of vertebra in the trunk region. They further study the levels of expression of Nr6a1 in single and compound mutants of miR196 and GDF11 embryos that have an increased number of trunk vertebrae and observe a clear correlation between the number of vertebrae and the levels of Nr6a1 expression which further supports the role of Nr6a1 in the trunk elongation process. They further find that Nr6a1 is sufficient to control the temporal activation of trunk Hox genes and prevents the expression of more posterior Hox genes. Finally, using embryonic stem cell differentiation experiments they analyse the role of Nr6a1 in the cell lineage choice of axial progenitors and identify that Nr6a1 favours the neural differentiation. Overall, the manuscript is clearly written and very well presented. The data provide compelling evidence that there are two distinct processes that control the trunk and tail modules of axial elongation and that Nr6a1 is a master regulator of the trunk growth. The reviewer is happy to see that the findings are in agreement with the temporal changes in the transcriptional signature of NMPs that is essential to support this very dynamic process during embryonic development.

I have only a few comments to bolster the findings of the paper regarding the role of Nr6a1 in axial elongation. These are summarized below:

- Nr6a1 is expressed in NMPs of wild type embryos at e8.5 and is downregulated at e9.5. However, it is not clear if the expression of Nr6a1 is maintained in the NMPs of the GDF^{-/-} for longer? The authors should include immunofluorescence data of Nr6a1 with T/Bra/Sox2 at e8.5, e9.5, e 10.5

Unfortunately, we have tested several commercial antibodies for Nr6a1 but are yet to find one that works in immunofluorescence. We have attempted to address this question by performing triple fluorescent staining for Nr6a1 transcript (fluorescent tyramide-amplification method), T/Bra protein and SOX2 protein in somite-matched E9.5 embryos. Unfortunately the high background observed for Nr6a1 using this method precluded a confident analysis.

We understand that the E10.5 *Gdf11*^{-/-} tailbud image in Figure 1D looks mainly neural and perhaps this leads to question whether NMPs are still present (and Nr6a1 within). Based on the work of Aires et al., (Dev Cell, 2018 – see Fig 1 and 2), there are still many Sox2⁺;T/Bra⁺ cells in the tailbud at this stage, and actually slightly more NMPs in *Gdf11*^{-/-} tailbud, so we do expect that *Nr6a1* will be maintained in the NMPs at this stage. But as we cannot experimentally validate this, we have kept our conservative assessment as follows:

Pg. 9. Spatial analysis of *Nr6a1* in E10.5 *Gdf11*^{-/-} embryos confirmed persistent expression in the caudal neural tube and mesoderm expanding toward the embryonic tip (Figure 1D).

- Is the expression of *Raldh2* and *Cyp26a1* affected in the posterior part of the *Nr6a1*^{-/-} embryos?

We have performed whole mount in situ hybridisation for both genes at E9.5 and E10.5, in somite-matched wildtype and *TCre*^{ERT2};*Nr6a1*^{flx/flx} embryos. We have included this new data as a supplementary figure (Figure S6) and modified the text as follows:

Pg. 11. We went on to assess the expression of retinoic acid signalling components *Aldh1a2* and *Cyp26a1* at E9.5 and E10.5, but see no major spatio-temporal differences between genotypes (Figure S6) that prefigure the rostral shift in *Hox* expression or that may underlie body plan changes.

We do want to note that *Aldh1a2* was identified as differentially expressed in *Cdx2P:Nr6a1* tailbuds and *Nr6a1*^{-/-} NMP differentiation, as part of the neural vs mesodermal gene expression analysis (Figure 8). This site of expression is different to the more commonly discussed *Aldh1a2* expression observed in somites, and we do not detect it by WM-ISH (Figure S6). In the context of the tailbud, *Aldh1a2* has been identified as a marker of MPs, tightly correlating with *Msg1* and *Tbx6* (Gouti et al, 2017), and this is consistent with our analysis in Figure 8. We have clarified this potential confusion as follows:
Pg. 20. (Figure 7A; Note: *Aldh1a2* in this context is a marker of MPs (Gouti et al., 2017)).

- The authors should include immunofluorescence data of T/Bra/Sox2 to prove the efficient generation of NMP cells at day 3 in the absence of Nr6a1.

This data is now included in Figure S11C.

- Have the authors compared the transcriptional signature of the Nr6a1^{-/-} d3 NMPs to the in vivo e8.5 and e9.5 NMPs? Are the Nr6a1^{-/-} NMPs more similar to e9.5 NMPs?

We have performed this analysis, and include the new data as supplementary figure (Figure S12C). Using the E8.5 and E9.5 in vivo NMP gene lists from Gouti et al., 2017, we see the temporal identity of Nr6a1-deficient d3 NMPs is reflective of an E8.5 NMP – and in fact, a statistically heightened E8.5 signature and concomitant reduced E9.5 signature. This data also matched the *Hox* expression analysis (Figure 7) where we see very little change in posterior *Hox* expression at Day 3 (with the exception of *Hoxb13*), and it is only after Gdf11 is added that the marked increase in posterior *Hox* become apparent in *Nr6a1*^{-/-} cells. We have included this in the text as follows:

Pg. 18. These *Nr6a1*^{-/-} ESCs were able to generate NMPs with equal kinetics to wildtype cells (Figure S11A-D), though whether the temporal progression of axial identity within these NMPs was altered, as may be predicted from the *in vivo* deletion of Nr6a1, remained unclear. To this end, we performed RNAseq analysis as differentiation proceeded, and first assessed expression of *Pou5f1* (*Oct4*), a known Nr6a1 target (Fuhrmann et al., 2001) and E8.5-NMP marker (Gouti et al., 2017). *Pou5f1* expression was slightly increased on d2 before the addition of Wnt, with a peak in enhanced differential expression observed on d3 (Figure S12B). We found this was accompanied by a statistical enrichment of many E8.5-NMP signature genes on d3, and concomitant reduction in E9.5-NMP signature genes (Figure S12C). This surprising result suggests that the deletion of Nr6a1 does not autonomously cause a speeding up of developmental timing within NMPs, but that additional factor(s) may be required for the earlier observations *in vivo*.

- Is the expression of Nr6a1 maintained longer in NMPs treated with TGFb inhibitor between d3 and d4?

Due to staff restrictions, we have not been able to perform this *in vitro* experiment, however *in vivo*, we have shown that *Nr6a1* expression is posteriorly expanded in *Gdf11*^{-/-} embryos mutants (Figure 1D).

Minor points

- In Figure 3B there are no red arrows that indicate the region containing the NMP cells as described in the legend.

Thank you, this has been corrected.

- In the title of Figure S9, there is a typo on ESCs

Thank you, this has been corrected.

Reviewer #3 (Remarks to the Author):

The paper by Chang et al. addresses the role of the Nuclear receptor subfamily 6 group A member 1 (NR6A1) in somitogenesis and trunk elongation. Nr6a1 is broadly expressed in the developing mouse embryo, including the neuromesodermal progenitors (NMPs). Single nucleotide polymorphisms (SNPs)

within Nr6a1 in mammals have been associated with increased trunk vertebral number. Based on these observations, the Authors hypothesize that Nr6a1 has a role in vertebrae formation in mice.

To address this question, they analyzed Nr6a1 expression via in situ hybridization in mouse embryos. They performed skeletal preparation analyses to assess the skeletal defects of the Nr6a1 loss of function and overexpressing alleles. They analyzed the paraxial mesoderm defects looking at the changes in somites identity by studying the Hox-code expression via RNAseq in the mutant embryos. Finally, the authors established an in vitro cell culture system to corroborate the findings observed in the developing embryos. The authors reported that Nr6a1 regulates posterior Hox genes expression in mouse tailbuds, and thus has a critical role in controlling the number of lumbar and sacral vertebrates. Additionally, they proposed that Nr6a1 plays an essential role in cell lineage choice of axial progenitors regulating the balance between neural and mesodermal fate. Finally, they suggested that Nr6a1 transcriptional regulation is mediated by the priming activity of SMAD2/3 at the epiblast state.

While the hypothesis that changes of vertebrae identity in Nr6a1 mutants is linked to the shift in somite identity and altered Hox code is quite intriguing, the experiments included in the current version manuscript do not fully support the claims and hypothesis.

The manuscript focused mainly on the regulation of Hox genes expression by NR6A1. HOX are critical key players of axial elongation and vertebrae patterning. However, the mechanism by which Nr6a1 controls posterior Hox genes is not fully elucidated.

Regarding the methodology employed, the authors generated two new mouse alleles, overexpressing Cdx2::Nr6a1 and new Nr6a1 mutant ES cell lines. However, no major cutting-edge technologies are used or implemented in the current manuscript.

In the opinion of this reviewer, there are no molecular methods included in this manuscript that shed light into an indirect or direct regulation of Hox genes by Nr6a1, yet it is a major claim in the paper. Dissecting the mechanisms by which Nr6a1 controls Hox expression is mandatory to understand the vertebrae phenotypes observed in the Nr6a1 mutants, and it is a key missing element of the paper.

We agree with the reviewer that we do not currently know the mechanism by which Nr6a1 controls Hox temporal progression - if we have inadvertently made any claim regarding this throughout the manuscript then this was made in error and will be corrected. However, this does not diminish the importance of identifying a completely novel regulator of *Hox* temporal progression, and our vertebral phenotypic assessment will hold true, whether or not it is ultimately shown that Nr6a1 physically binds Hox loci (or not). I would like to reiterate the key advance of our work within the context of the field: We can exquisitely alter A-P axis length, positively and negatively, based on the level of Nr6a1. This is a key result that to the best of my knowledge has never been achieved before, even for fundamental developmental molecules such as Wnt, RA, or Fgf.

Major critiques:

The authors reported that the expression and regulation of Nr6a1 correlate with trunk formation in the mouse. Although they said temporal enrichment in NMPs and somites between E9.5 and E10.5, Nr6a1 appears to be broadly expressed, including in the epiblast stages. What provides specificity to its function in NMPs and somites? The authors should use immunofluorescence (IF) to assess NR6A1 protein activity in the tailbud and test whether it has specific nuclear staining in the caudal lateral epiblast, NMP territories, or somites.

We have tested numerous commercial antibodies against Nr6a1 but unfortunately none work in whole mount or section immunofluorescence. To circumvent this, we interrogated a single-cell RNAseq dataset to show that while *Nr6a1* transcript levels are indeed broad as judged by RNA in situ hybridisation (Fig 1A), *Nra6a1* is enriched in NMPs, caudal mesoderm and caudal epiblast, compared with other cells types such (Fig 1B). However, we did not intend to suggest that the function of *Nra61* is restricted to the NMPs, and it is possible that it Nr6a1 functions in other progenitor populations of the tailbud or beyond. We have clarified this in the text as follows:

Pg.8. Interrogation of a published single cell RNA-seq dataset produced from E6.5 and E8.5 embryos (Pijuan-Sala et al., 2019) confirmed robust expression of Nr6a1 at these early stages in NMPs, caudal

mesoderm and the caudolateral epiblast (Figure 1B), with lower expression in many other progenitor and mature populations where it has the potential to function.

The sentence: “these data revealed a dynamic pattern of Nr6a1 expression that correlates with axial progenitors of the trunk region and one that is terminated by the synergistic action of miR-196 and Gdf11 signaling” should be clarified.

We agree that we should not have used the term axial progenitors, as it was whole tailbud tissue that was collected for this experiment, and have clarified the sentence as follows:

Pg. 9. Collectively, these data revealed a dynamic pattern of Nr6a1 expression within tailbud tissue over time, with expression terminated by the synergistic action of miR-196 and Gdf11 signaling.

Overall it is not clear the rationale for the experiments in Fig 1C-D. What is the author’s intention? Do they want to study the genetic interaction between Nr6a1 and Gdf11? If this is the case, why? And why is it addressed at this point before knowing if Nr6a1 has a role in axial elongation?

In recently published work from our lab (Hauswirth et al., Nat Comm 2022), we identified a novel interaction between Gdf11, miR-196 and retinoic acid- and were able to increase thoraco-lumbar number in a graded manner based on the levels of these various factors. The in situ hybridisation data of Fig 1A suggested that Nr6a1 may act specifically during trunk formation as it was extinguished in the tailbud at later stages. We reasoned that if Nr6a1 had a functional role in trunk formation, then its expression should be maintained in the tailbud of embryos which have an expanded thoraco-lumbar count. The clear correlation between Nr6a1 expression level and the degree of thoraco-lumbar expansion of the various mutants supported further investigation of this gene during axial elongation.

The authors should revise the manuscript and condense the panels of Fig1 and 2 in one Figure. The main message of Fig1 should be the expression and phenotype of the mutant Nr6a1 embryos.

We would prefer to keep the two figures separate as the data are addressing separate points and may be distracting as one large panel. We are happy to discuss further if this is required editorially.

While it is clear that there are different numbers of lumbar and sacral vertebrae, it is not so clear if this phenotype is a result of a change in somite identity or an aberrant number of somites (Fig2F). The somites should be counted using reference points, both the forelimb and hindlimb (like in McPerson et al. 1999). A table providing statistical significance of the observed difference between controls and mutants should be included.

Somite number was counted relative to the hindlimb landmark - this was displayed in the right panel of Fig2F (now Fig 3B). However, as somite morphology within the interlimb region was quite aberrant, as was morphology of the early hindlimb bud, we had chosen to quantify segment number by counting total vertebral number (TVN) at E16.5.

Supplementary Figure 3A, left panels, shows total vertebral number following allelic deletion of Nr6a1, with statistical analysis presented in the lower panel. In the text, we described the analysis of this data as follows:

Pg 11. heterozygous conditional deletion of Nr6a1 resulted in 2 less T elements (Figure 2A) and an overall reduction in TVN by 1 (Figure S3A).... This phenotype was enhanced following homozygous conditional deletion of Nr6a1, with the number of sternal rib attachments reduced to 5, an overall loss of 4 T elements and a reduction in TVN by 3 (Figure 2A and S3A).

The subsequent analysis of specific changes in thoraco-lumbar and sacro-caudal counts can be found in Figure 2D, with statistical analysis in the lower panel of this Figure.

The authors claimed that Nr6a1 inhibits posterior Hox gene expression in the tailbud. There is a differential expression of Hox genes, but that does not necessarily mean that Nr6a1 inhibits Hox expression in the tailbud.

We agree that our use of the work inhibits was incorrect, and not the way we view the function of Nr6a1 which we see as an important regulator of timely posterior Hox activation. We have revised two subsection titles as follows:

Pg. 13. Nr6a1 controls timely posterior Hox gene activation in the tailbud

Pg. 38. Figure 2. Nr6a1 is required for trunk elongation and timely activation of sacro-caudal identity

The effect on the Hox could be a result of a general developmental delay.
Please see our detailed response to Reviewer 1 who raised a similar point.

To prove that Nr6a1 inhibits posterior Hox gene expression, the underlying mechanism of inhibition should be clearly addressed.

In other words, how does Nr6a1 regulate Hox genes in the tailbuds? Are the 4 Hox paralogs the direct targets of NR6A1? ChIP addressing the direct targets of NR6A1 using either embryonic tissues or cultured cells is critical and mandatory to address this point.

In addition to the Hox genes, which other genes are differentially regulated in the Nr6a1 mutant tailbuds? Maybe the effect on Hox genes is indirect and Nr6a1 controls important upstream regulators like WNT.

Finally, what provides specificity to a broadly expressed gene like NR6A1 in the tailbud. How can it target specifically the posterior Hox genes?

The authors should consider generating a NR6A1-GFP or TAG allele, valid for the ChIP experiments (if a ChIP grade antibody is not available), and immunoprecipitate NR6A1 to identify potential binding partners that could provide information about the DNA-binding specificity.

These are all excellent suggestions and constitute ongoing experiments within our lab, but are beyond the scope of this manuscript.

Regarding the question "In addition to the Hox genes, which other genes are differentially regulated in the Nr6a1 mutant tailbuds?"

We thank the reviewer for highlighting that we did not upload the full list of differentially expressed transcripts identified within *Cdx2P:Nr6a1* overexpressing tailbuds. This has been corrected, please see Supplementary Table 2. We have compared this dataset with a similar dataset generated from *Gdf11*^{-/-} tailbuds, revealing a 151 common gene list (now included in Supplementary Table 2) which includes not only Hox genes, but many NMP-enriched genes and one gene that has been experimentally validated to control vertebral number (*Lin28a*). The description of this analysis can be found on pg.16.

The authors hypothesized that the panel of Hox genes expression in Fig 3A is associated with the NMPs. Still, Hox genes are expressed in many other cell types, including neurons. Colocalization with NMP markers or the use of the scRNA-seq is required to unequivocally attribute the precocious expression of posterior Hox genes to NMPs. A similar consideration can also be raised for Fig 3B.

Performing scRNAseq analysis will significantly help with the attribution of Hox genes to the NMP cluster and dissect the phenotypes.

For Figure 3A, we believe we only use the term tailbud, but are happy to correct if we have used NMP. For Figure 3B, we agree that without colocalisation analysis, we cannot infer upregulation in regions harbouring NMPs and have removed any reference to NMPs within the results section relating to Figure 3B.

Pg. 13. In contrast, E9.5 *TCre^{ERT2};Nr6a1^{flx/flx}* embryos showed additional ectopic expression in the tailbud mesoderm (Figure 3B). Ectopic *Hoxb13* expression persisted in E10.5 *TCre^{ERT2};Nr6a1^{flx/flx}* embryos throughout the tailbud mesoderm including the NMP-containing chordoneural hinge, and increased *Hoxb13* expression levels were observed in the neural tube relative to wildtype (Figure 3B).

To test if Nr6a1 clearance is essential for the trunk-to-tail transition, the authors generated a new mouse allele where Nr6a1 is under the control of the *Cdx2* promoter. So that Nr6a1 will be expressed at E10.5 in the tailbud when Nr6a1 is no longer present. *Cdx2::Nr6a1* showed an increased number of thoracolumbar vertebrae.

If NR6A1 controls posterior Hox gene expression, why the thoracic vertebrae that are controlled by middle Hox are affected? This observation raises again the question of whether the Hox genes are the primary targets. The phenotype of the thoracic vertebrae is strong while the deregulation of thoracolumbar Hox genes are modest. So what does NR6A1 really do?

We have addressed these points in the text as follows:

Pg. 15. At a morphological level, we do not believe the minor increase in trunk *Hox* expression seen in *Cdx2P:Nr6a1* tailbuds (Figure 5C) is the cause of the expanded thoraco-lumbar region, but rather, it is the delay in overall trunk-to-tail *Hox* code transition. Simply maintaining a trunk *Hox* code, in the absence of posterior *Hox* activation, would suffice. At a molecular level, it is possible that the minor increase in trunk *Hox* expression may be a secondary effect. In WT tailbuds, the timely activation of a posterior *Hox* code may feedback and reduce trunk *Hox* expression levels, a scenario that is not occurring at the same time in *Cdx2P:Nr6a1* tailbuds.

The authors claim that “Nr6a1 alone is sufficient to control the temporal progression of *Hox* activation of all 4 *Hox* clusters.” – this sentence should be toned down because a direct link between Nr6a1 and *Hox* gene regulation has to be demonstrated. Furthermore, NR6A1 cannot work alone, but it has to work together with cell-specific factors. Otherwise, a similar effect on *Hox* genes should be seen in all the territories where Nr6a1 is expressed.

We have taken out the word alone, as we did not intend to suggest that the Nr6a1 protein acts alone (which as the reviewer points out is unlikely the case for a nuclear receptor). We have adjusted the text as follows:

Pg.15. In summary, we show that Nr6a1, likely acting with as-yet unknown protein partners, is sufficient to control the temporal progression of *Hox* activation of all 4 *Hox* clusters

In the differentiation system (Fig6), the authors do not consider the cell populations' heterogeneity during differentiation. At d2, they listed the generation of EpiSCs; what is the evidence that those are EpiSCs? Or that at d3 there are NMPs? The only markers used to attribute cell identity are the *Hox* genes. As discussed in Figure 3, *Hox* genes are expressed in many other cell types. Colocalization with NMP markers or the use of the scRNA-seq is required to unequivocally attribute the precocious expression of posterior *Hox* genes to NMPs.

We agree we should not have omitted the characterisation performed that confirmed our *in vitro* differentiation protocol was behaving as previously published. This has now been included in Supplementary Figure S8. Transcriptomic analysis revealed the transition from d2 (epiblast-like) to d3 (NMP-like) was marked by loss of EpiSC markers *Otx2* and *Fgf8*, and the increase of NMP marker genes *Cdx2*, *Cdx4*, *T/Bra* and *Nkx1.2*. Protein analysis confirmed at a cellular level Sox2⁺;T/Bra⁺ dual positivity across the majority of the NMP colonies. The E8.5-NMP signature was reduced at d4 compared with d3 as expected, and the E9.5-NMP signature robustly activated following addition of Gdf11 as expected. scRNAseq has previously confirmed the expression of *Hox* genes within *in vitro* NMPs derived under conditions such as those used here (Gouti et al., 2017).

What do you expect to find at d4?

The data for d4, with or without Gdf11, was presented in original Figure 6 (now Figure 7).

The authors suggested that Nr6a1 promotes neural fate within bipotent NMPs. Could a neural phenotype be identified in the Nr6a1 mutant embryos? Is the neural tube duplicated like in the Tbx6 mutants?

Where are these progenitors localized in the embryos?

NMPs fated to neuronal and mesodermal fate should be identified in the embryos using specific lineage markers. Neuronal and mesodermal progenitors should be counted and compared to the controls.

Please see our detailed response to Reviewer 1, but in short, we do not see dramatic changes such as duplicated neural tube, however we have counted NP and MP within the tailbud as suggested, and we see a significant increase in MP and decrease in NP cell number in *TCre^{ERT2};Nr6a1^{flx/flx}* embryos (Figure 8C).

In Fig 7, why the comparison between *in vivo* and *in vitro* systems are done with two different mutated alleles?

We wanted to assess both gain- and loss-of-function scenarios. *In vitro*, we only have a loss of function tool, and thus we compared this with the RNAseq dataset from *in vivo* Nr6a1 gain of function.

The authors suggested that Nr6a1 is regulated by its antisense transcript, showed by the opposite

pattern of expression during differentiation. They proposed a potential priming effect of SMAD2/3 in the epiblast that regulates Nr6a1^{los} and, in turn, Nr6a1. Given these observations are purely speculative, they should mention them in the Discussion section only.

We are happy to move this section here, and we check editorially if this appropriate given there is some data presentation in this section.

To prove that Nr6a1^{los} regulates Nr6a1 via SMAD2/3. The authors should delete the SMADs binding sites of Nr6a1^{los}, see the effect on Nr6a1 expression, and test if it recapitulates the phenotype?

We thank the reviewer for the constructive suggestion to be investigated in future.

No precise details about the statistical methods used have been reported. This information is also missing in the Material and Methods.

This information can be found under the header - Quantification and statistical analysis - pg 58-59, and throughout the figure legends.

Western blot proving the absence of NR6A1 in ES cell lines should be provided. Given the mutant allele has been generated by deleting the DNA binding motif, a truncated protein can be produced, raising the possibility of a dominant-negative effect.

We appreciate this comment, but in the absence of a working antibody this is currently not possible.

The Discussion is nicely written, but it sounds as Evo-devo review and it is unlinked with the findings described in the Results.

We respectively disagree. We do not mention an evolutionary link until the final paragraph, and believe the discussion is focused on interpreting the cellular and molecular discoveries made.

REVIEWERS' COMMENTS

Reviewer #1 (Remarks to the Author):

Overall the manuscript is much improved by the addition of further data and modifications to the text. Descriptions of the phenotypes are clearer and the addition of Hox in situ hybridisation further strengthens the case that Nr6a1 perturbation leads to lumbar-sacral transformations.

In general my points are satisfactorily answered and I have only the following remaining minor comments:

.pg 15 the second inserted section beginning 'at a morphological level...' is now quite speculative and I am not sure whether it adds much to the overall argument. It could be cut, or possibly the points could be better made if moved to the discussion.

The intracellular FACS presented in Fig 8 and S13 are incompletely described- controls for gating and number of repeats are required to be confident about the rather minimal effects observed.

Figure S12B shows Nr6a1 is elevated versus wildtype in Nr6a1^{-/-} cells versus wildtype at day 3 of differentiation in vitro. Can the authors clarify this? Presumably it indicates an excess of mutated Nr6a1 transcript? how sure are the authors that the ESC mutation is null?

Reviewer #2 (Remarks to the Author):

The authors have addressed my concerns in the revised version of the manuscript.

Reviewer #3 (Remarks to the Author):

This reviewer appreciates the effort of the Authors to answer the questions raised, in particular the revision of Fig1 with the ScRNAseq analysis and the overall revision of the manuscript.

Dissecting the mechanism by which Nr6a1 controls Hox expression is critical to understand the phenotypes observed in the Nr6a1 mutants. I still think that this aspect should address more deeply, and I hope to see the follow-up of this story soon.

I think it should tone down the statement that Nr6a1 can exquisitely alter the A-P axis length. The HOX cofactors (TALE proteins) can also modulate A-P axial patterning.

Overall I find the author's revision satisfactory.

Reviewer #1 (Remarks to the Author):

Overall the manuscript is much improved by the addition of further data and modifications to the text. Descriptions of the phenotypes are clearer and the addition of Hox in situ hybridisation further strengthens the case that Nr6a1 perturbation leads to lumbar-sacral transformations.

In general my points are satisfactorily answered and I have only the following remaining minor comments:

.pg 15 the second inserted section beginning 'at a morphological level...' is now quite speculative and I am not sure whether it adds much to the overall argument. It could be cut, or possibly the points could be better made if moved to the discussion.

We have moved these sentences to the discussion, please see inserted text on pg. 25 with minor modifications for clarity in this new context.

The intracellular FACS presented in Fig 8 and S13 are incompletely described- controls for gating and number of repeats are required to be confident about the rather minimal effects observed.

We apologise for this omission. Embryo replicate number has been added in the main text as follows:

Pg. 21. WT n=4 and CKO n=9 individual tailbuds from across two litters
This information is also now included in the figure legend on pg. 43.

Details of cell viability stain have been added to the methods section as follows:

Pg 62. ...resuspended in PBS + 2%FBS and 1 µg/ml Propidium iodide

Details of controls for gating have been added to the methods section and Supp Fig 13 as follows:

Pg 62. To set gates, all non-tailbud tissue was pooled and stained for 1) both secondary antibodies, 2) anti-Sox2 primary and secondary alone and 3) anti-T/Bra primary and secondary alone. Experimental cells were first gated to remove debris, then gated based on size to select for single cells, and finally, cells that had taken up Propidium iodide were excluded leaving single viable cells for analysis of Sox2 and T/Bra expression as depicted in Supplementary Figure S13.

Figure S13 updated to include controls, and Figure S13 Legend altered as follows:

Pg.51. **Figure S13. Flow cytometry gating parameters**

A. Control staining used to establish gating parameters.

B. Representative plots depicting final gating parameters for experimental samples. Cell suspensions for analysis were gated to remove debris, isolate single cells, remove non-viable cells before detection of Sox2 and T/Bra secondary fluorophores.

Figure S12B shows Nr6a1 is elevated versus wildtype in Nr6a1^{-/-} cells versus wildtype at day 3 of differentiation in vitro. Can the authors clarify this? Presumably it indicates an excess of mutated Nr6a1 transcript? how sure are the authors that the ESC mutation is null?

The reviewer is correct, and we interpret this as the null cell positively feeding back onto the locus in an attempt to rectify the loss of Nr6a1 activity. We cannot formally conclude the ESC mutation is a null and thus have altered the text to be clear exactly what the two mutant alleles generate and our expectations for this loss-of-function line.

Pg. 49. The alignments of wildtype *Nr6a1* with $\Delta 219$ and $\Delta 125$ *Nr6a1*. The $\Delta 125$ mutant allele causes a premature truncation within the DNA binding domain, removing 369 amino acids relative to the wildtype protein. The $\Delta 219$ allele causes a 73 amino acid deletion, removing almost the entirety of the DNA binding domain. The similar molecular outcomes observed for *in vitro* and *in vivo* loss-of-function datasets supports this being a loss-of-function ESC line, though this remains to be formally proven.

Reviewer #2 (Remarks to the Author):

The authors have addressed my concerns in the revised version of the manuscript.

Reviewer #3 (Remarks to the Author):

This reviewer appreciates the effort of the Authors to answer the questions raised, in particular the revision of Fig1 with the ScRNAseq analysis and the overall revision of the manuscript.

We thank the reviewer for these positive comments.

Dissecting the mechanism by which Nr6a1 controls Hox expression is critical to understand the phenotypes observed in the Nr6a1 mutants. I still think that this aspect should address more deeply, and I hope to see the follow-up of this story soon.

I think it should tone down the statement that Nr6a1 can exquisitely alter the A-P axis length. The HOX cofactors (TALE proteins) can also modulate A-P axial patterning.

I remember writing the sentence “exquisitely alters A-P axis length” but I can no longer find the text anywhere in the manuscript using the search function. I am happy to change if it is still there. I agree with Reviewer #3 that there are many factors that modulate A-P patterning, though whether they alter A-P axis length (i.e. vertebral number) is still unknown.

Overall I find the author’s revision satisfactory.

Additional changes not requested but altered to improve manuscript

- Italicised any gene name that was missed
- Title page, corrected spelling errors: Medical, Kansas
- Deleted “two” on pg 5 – relevant to the original comment by Reviewer#1 that there are likely more than two developmental modules controlling vertebral column formation.
- Added clarification on pg 15: “Growing evidence however has challenged this dogma (Young et al., 2009; Hauswirth et al., 2022) supporting the view that the observed altered *Hox* code may indeed drive meristic changes.”
- Updated Data deposition section on Pg. 53 to include: Original data relating to the the microarray experiment can also be accessed from the Stowers Original Data Repository <https://www.stowers.org/research/publications/LIBPB-1647>
- Added more information to Supp Figure 2C legend as follows: Pg. 46. Shades of red represent experimental animals and shades of blue represent control animals. Error bars represent the mean with standard deviation. A two-tailed t-test was used for statistical comparisons between experimental and control embryos under the same treatment
- Updated **Quantification and statistical analysis** section to include CMVCre analysis as follows:
Pg. 63

Vertebral formulae - *CMVCre*^{ERT2} analysis

For statistical analysis and data visualisation of vertebral numbers, Graph Pad Prism 9.3.1 (471) was used. A two-tailed T-test was employed. Error bars represent the mean with standard deviation.

Vertebral formulae - *TCre*^{ERT2} analysis